# Genomic signatures of recent convergent transitions to social life in spiders

Chao Tong [1,2] ✉, Leticia Avilés [3], Linda S. Rayor [4], Alexander S. Mikheyev[5] & Timothy A. Linksvayer [1,2] ✉

The transition from solitary to social life is a major phenotypic innovation, but its genetic underpinnings are largely unknown. To identify genomic changes associated with this transition, we compare the genomes of 22 spider species representing eight recent and independent origins of sociality. Hundreds of genes tend to experience shifts in selection during the repeated transition to social life. These genes are associated with several key functions, such as neurogenesis, behavior, and metabolism, and include genes that previously have been implicated in animal social behavior and human behavioral disorders. In addition, social species have elevated genome-wide rates of molecular evolution associated with relaxed selection caused by reduced effective population size. Altogether, our study provides unprecedented insights into the genomic signatures of social evolution and the specific genetic changes that repeatedly underpin the evolution of sociality. Our study also highlights the heretofore unappreciated potential of transcriptomics using ethanol-preserved specimens for comparative genomics and phylotranscriptomics.

The evolution of sociality is a phenotypic innovation that occurred repeatedly and sporadically across vertebrates, insects, spiders, and crustaceans[1]. Sociality and social behavior, more generally, are hypothesized to evolve via changes in a set of deeply conserved genes[2–6]. Many transcriptomic studies have emphasized overlapping sets of genes or molecular pathways underlying the expression of key social phenotypes in different lineages[7–10]; others emphasize the importance of lineage-specific genes, or genes with lineage-specific expression[11–13]. Studies searching for patterns of molecular evolution associated with the convergent evolution of sociality have also yielded differing results. For example, a population genomic study of two bee and one wasp species, altogether representing two origins of sociality, found significant overlap in genes experiencing positive selection among all three species[14]. In contrast, the largest comparative genomic study of sociality to date, which included ten bee species representing two independent origins and two independent elaborations of eusociality, found low and insignificant overlap for rapidly evolving genes associated with social evolution in each lineage[15]. Overall, the degree to which convergent social evolution in diverse lineages involves consistent changes in the same or similar genes remains largely unclear.

One difficulty in identifying genomic signatures of sociality is the relatively few and often ancient origins of this trait[16]. Social hymenopteran insects (wasps, bees, and ants), and bees in particular, for instance, are often described as ideal study systems because they include the full range of social complexity and multiple origins of sociality[15,17]. However, there are only an estimated 2–3 independent origins of sociality within the bees, and 7–8 across all hymenopterans, with these origins occurring between 20 and 150 million years ago[15–17]. Thus, other lineages with many independent and more recent origins of sociality can complement studies in social insects and help to further identify common genomic signatures of social evolution.

[1]Department of Biology, University of Pennsylvania, Philadelphia, PA 19104, USA. [2]Department of Biological Sciences, Texas Tech University, Lubbock, TX 79409, USA. [3]Department of Zoology, University of British Columbia, Vancouver, British Columbia V6T 1Z4, Canada. [4]Department of Entomology, Cornell University, Ithaca, NY 14853, USA. [5]Evolutionary Genomics Group, Research School of Biology, Australian National University, Canberra 0200, Australia. ✉e-mail: tongchao1990@gmail.com; tlinksvayer@gmail.com

Like the social insects, spiders exhibit a wide range of social complexity and multiple independent origins of sociality[18–20]. Sociality evolved independently in spiders an estimated 15–16 times[21,22], with each independent origin thought to be relatively recent, at most a few million years ago[23,24]. Spiders are classified into several categories of social organization, including solitary, subsocial, prolonged subsocial, and social: solitary species live in individual nests, having dispersed from the egg sac soon after hatching[18,22]; subsocial spiders have nests that contain a single mother and up to a few dozen offspring, which may remain together for several instars before dispersing to initiate their own nests[18,19]; prolonged subsocial spiders form colonies containing a single mother and multiple cohorts of offspring, which remain in their natal nest until late adolescence or sexual maturity before dispersing to independently found their own colony[19]; and social spiders (i.e., non-territorial permanent-social species) form colonies that contain multiple adult females and offspring that remain in the natal nest through maturity, mating with each other to produce new generations that reoccupy the natal nest[18]. In social and subsocial species, individuals cooperate in building and maintaining the communal nest, capture prey cooperatively, and share their food. In social species, individuals also exhibit communal brood care. Social spider species have additional distinct features, including colonies that grow through internal recruitment over multiple generations, in some species reaching sizes of tens of thousands of individuals, female-biased sex ratios, high reproductive skew, and high rates of inbreeding[20,25].

Spider sociality has a twiggy phylogenetic distribution, where the closest relatives of most social species are not social[21–24]. By comparing pairs of social species and their closest nonsocial (i.e., subsocial) relatives within a single genus, recent studies have begun to identify the genetic consequences of spider sociality and associated shifts from outbreeding to inbreeding[23,26–29]. These studies, together with a preliminary comparative genomic study of two social species from one genus and several solitary species[30], have begun to identify putative genomic signatures of the evolution of sociality in spiders, including elevated genome-wide rates of molecular evolution[26–30].

Here, we use comparative genomic approaches to determine whether there are statistically supported common genomic signatures associated with the repeated transition to sociality in spiders (Fig. 1). We use 22 spider species of a range of social systems and representing eight independent origins of sociality[24,31–33]. We test whether the convergent evolution of sociality across these lineages is associated with (1) convergent genome-wide patterns of molecular evolution; (2) convergent shifts in evolutionary rates for specific genes; and (3) convergent amino acid substitutions in specific genes. We show that genome-wide, gene-wide, and site-specific changes repeatedly occurred during recent convergent transitions to sociality in spiders. Altogether, our study provides unprecedented insights into the genomic signatures of social evolution and the specific genetic changes that repeatedly underpin the evolution of sociality.

## Results

### Spider-omics assembly, annotation, and phylogeny

We sequenced and assembled transcriptomes of 16 species in the genera *Theridion, Anelosimus* (family: Theriidiade), and members of the family Sparassidae (Fig. 1, Figs. S1 and S2, Supplementary Table 1). We complemented these new transcriptomes with six existing transcriptomes from closely related species[28,33–38] (Supplementary Table 2). In addition, we included two available genomes of social spider species, *Stegodyphus dumicola* and *Stegodyphus mimosarum*. For analyses that required an outgroup (i.e., the RERconverge analysis), we also included a solitary outgroup species, *Acanthoscurria geniculata* (family: Theraphosidae) (Supplementary Table 2). Moreover, we reassembled and improved the subsocial spider *Anelosimus studiosus* genome (GCA_008297655.1) with our newly sequenced *A. studiosus* transcriptome sequencing reads and annotated spider protein datasets (Supplementary Table 2). Finally, we assessed the completeness of the genome or transcriptome assembly using the BUSCO pipeline[39] (Fig. 1b), detecting an average of 91.24% arachnid conserved genes (2934 genes) in each spider species (Supplementary Table 3).

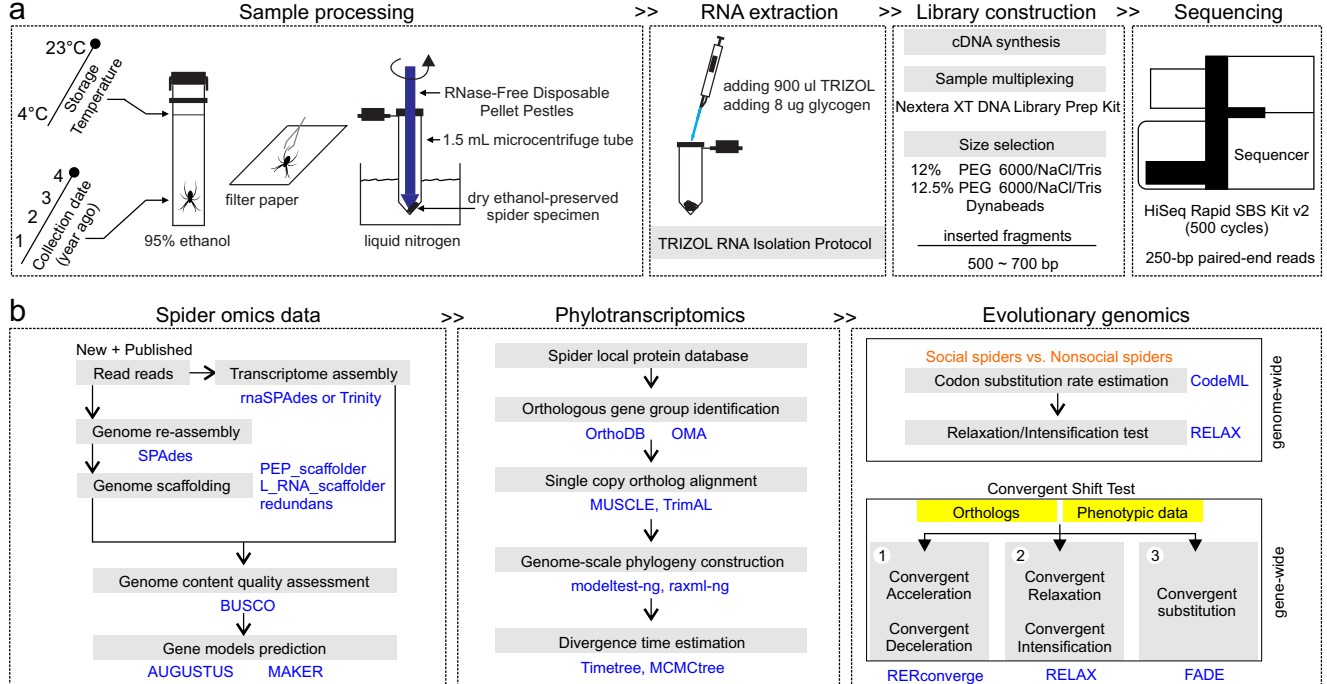

**Fig. 1 | The flowchart represents the analysis pipeline. a** Sample processing and transcriptome sequencing; **b** spider transcriptome and genome assembly, annotation, phylotranscriptomic analyses, comparative genomic analyses and evolutionary analyses. Technical details of the workflow are provided in the "Methods" section.

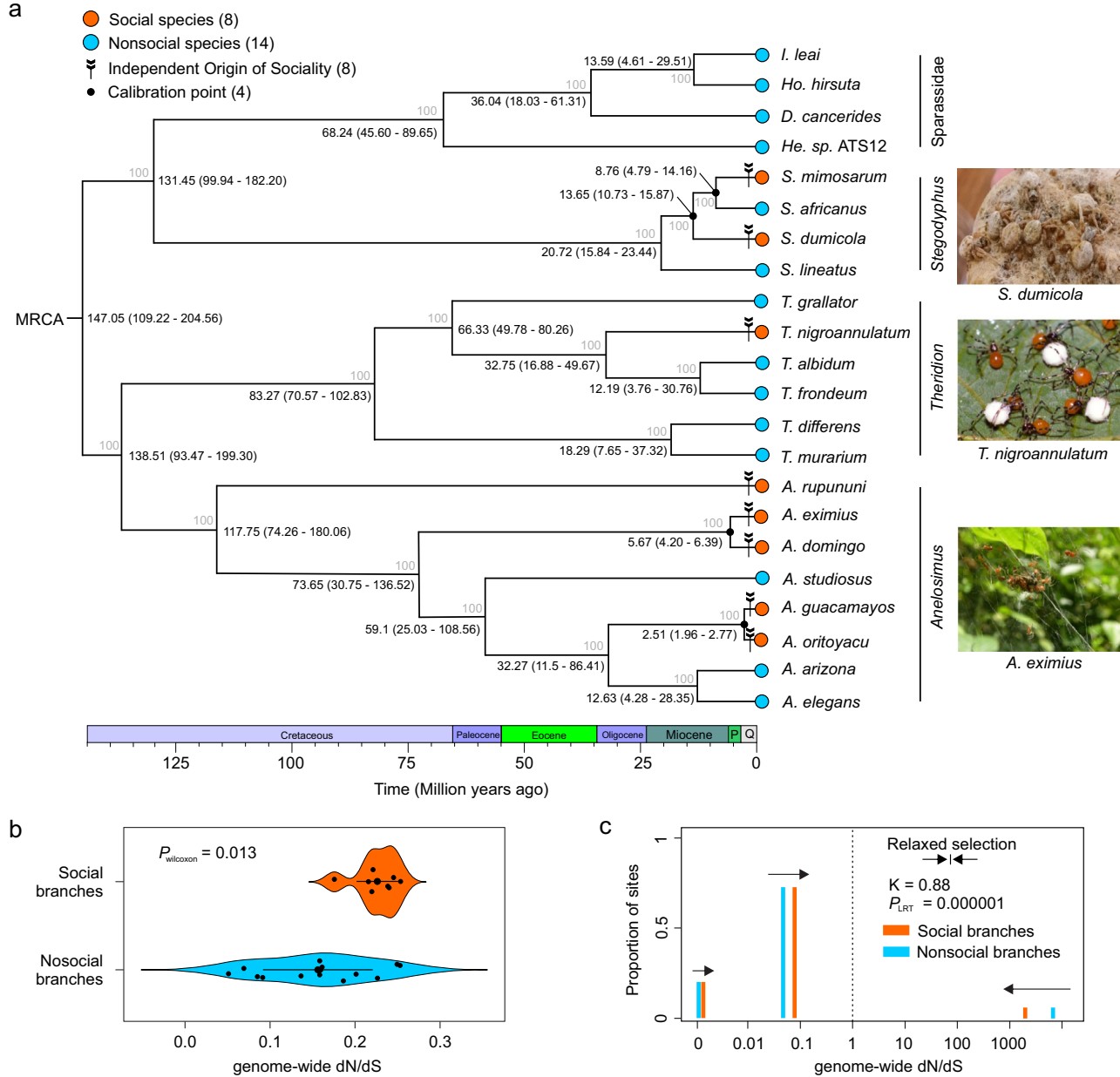

**Fig. 2 | Phylogeny of study species and genome-wide pattern of molecular evolution in social and nonsocial branches. a** The maximum likelihood (ML) phylogenetic tree with estimated divergence time of the 22 spider species included in the study. The ML tree was inferred from 3832 core-shared single-copy orthologs. Bootstrap values are indicated along the branches. The divergence times at the nodes were estimated using four calibrations indicated with solid black dots. Median age estimates and 95% highest posterior densities (Mya) are shown for each node. Q and P represent the Quaternary Period and the Pliocene Epoch, respectively. Four lineages of spiders are distinguished by dark columns: genera *Anelosimus*, *Theridion*, *Stegodyphus* and the family Sparassidae. Orange dots represent social spider species, light blue dots represent nonsocial species including prolonged subsocial, subsocial, and solitary spider species, the arrow tails represent the independent origins of sociality in spiders. Pictures of social spiders (from top to bottom) from *Stegodyphus dumicola* (photo credit: Noa Pinter-Wollman), *Theridion nigroannulatum* and *Anelosimus eximius* (photo credit: Leticia Avilés). **b** Violin plot depicting the genome-wide pattern of molecular evolution (dN/dS) between social ($n = 8$) and nonsocial spider ($n = 14$) branches across the phylogeny (horizontal bars indicate 95% CI of the means). *P* value was calculated by using Wilcoxon rank-sum test. Social spiders experienced convergent elevated genome-wide molecular evolution during the transition to social life. **c** Multi-bar plot depicting the patterns of selection experienced across sites in the genomes of social and nonsocial spiders, estimated with the Partitioned Descriptive model in RELAX. *P* value as calculated by using Likelihood-ratio test (LRT). The distribution of dN/dS across sites in the genomes are illustrated by three categories of dN/dS for social (orange) and nonsocial (light blue) branches. The vertical dashed line at dN/dS = 1 represents neutral evolution, bars at dN/dS > 1 represent sites experiencing positive selection, and bars at dN/dS < 1 represent sites experiencing purifying selection. The arrows show the direction of change in dN/dS between nonsocial and social branches, demonstrating relaxation of purifying and positive selection associated with the transition to sociality. K < 1 indicates genome-wide relaxation of selection.

We employed a combination of de novo and homology-based approaches to annotate the gene models of the assembled transcriptomes and improved reassembled genome, and obtained annotated gene models ranging from 19,224 to 96,176 per species (Supplementary Table 3). Then we used a reciprocal hit search strategy with OMA (www.omabrowser.org) and OrthoDB (www.orthodb.org), and identified a total of 7590 orthologous groups (OGs) and 3832 core single-copy orthologs that included all species (Supplementary Table 3).

Based on the concatenated single-copy ortholog data, we reconstructed the phylogeny for our study species (Fig. 2a), which was

strongly supported (bootstrap value = 100) for all nodes, and was mostly consistent with previously-published phylogenies[24,31–33,40] (Fig. 2a). We estimated divergence times for each phylogenetic tree node (Fig. 2a).

Notably, the 22 spider species included in all of our analyses spanned the full range of spider sociality: eight social species, one prolonged subsocial species, five subsocial species, and eight solitary species. Since we were interested in identifying genomic changes associated with sociality, we compared the eight social species (hereafter labeled as "social") with the 14 species in the remaining categories (hereafter all grouped together as "nonsocial") (Fig. 2a).

### Social spiders exhibit convergent genome-wide patterns of accelerated molecular evolution that is caused mainly by the relaxation of selection

As a first test of signatures of convergent molecular evolution, we asked whether social branches had different rates of genome-wide molecular evolution (i.e., dN/dS) compared to nonsocial branches. We estimated the genome-wide ratio of non-synonymous to synonymous substitution (dN/dS) across the spider phylogeny with the concatenated 3,832 single-copy ortholog dataset (Fig. 2b). We found that social spiders experienced higher genome-wide rates of molecular evolution compared to their nonsocial relatives, estimated by both CodeML in PAML 4.7a[41] (Wilcoxon rank-sum test, $p$-value = 0.013, Fig. 2b) and HyPhy 2.5[42] (Likelihood Ratio Test, LRT, $p$-value <0.00001).

Elevated dN/dS can be caused by increased positive selection, relaxed purifying selection, or a combination of both. RELAX[43] quantifies the degree to which shifts in the distribution of dN/dS across individual genes or whole genomes are caused by overall *relaxation* of selection (i.e., weakening of both purifying selection and positive selection, towards neutrality) versus overall *intensification* of selection (i.e., strengthening of both purifying selection and positive selection, away from neutrality). Specifically, RELAX models the distribution of three categories of dN/dS−positive selection, neutral evolution, purifying selection−across a phylogeny, comparing foreground (i.e., social) to background (i.e., nonsocial) branches and estimating a parameter K that indicates overall relaxation (K < 1) or intensification (K > 1). We found that the shifts in genome-wide dN/dS between social and nonsocial branches were caused mainly by relaxation of both purifying and positive selection (K = 0.88, LRT, $p$-value <0.001) (Fig. 2c).

### Hundreds of genes experience convergent shifts in gene-wide rates of molecular evolution in social spiders

Next we asked if there was evidence for convergent molecular evolution at the gene level. First, we applied the relative evolutionary rate (RER) test implemented in RERconverge[44] that is specifically designed to test for signatures of convergent molecular evolution underlying convergent phenotypic evolution (Fig. 1b). This test examines the rate of protein sequence evolution for each gene on every branch across the phylogeny, standardized to the distribution of rates across all genes. Genes for which these standardized rates (RER) are consistently either higher or lower in foreground branches (i.e., social branches) compared to background branches (i.e., all remaining branches) are identified as having experienced accelerated or decelerated molecular evolution in social species. We used a phylogenetically restricted permutation strategy, dubbed "permulations", to assess the statistical significance of genes and enriched GO terms. This permulation approach combines phylogenetic simulations and permutations to construct null expectations for $p$-values, and has been shown to be unbiased and more conservative than other approaches such as permutations alone[45,46]. Importantly, permulations also correct for non-independence among genes for GO term enrichment, which is not true for the nominal parametric $p$-values reported from RERconverge[44,47].

We ran 10,000 permulations and identified genes and enriched GO terms that experienced significant convergent shifts in RERs in social compared to nonsocial branches. The resulting gene and GO

term permulation $p$-values were corrected for multiple comparisons by computing $q$-values[48], and we considered $q$-values <0.15 (i.e., genes and GO terms with an estimated FDR < 0.15) to be significant. Out of 7590 single-copy orthologs included in REconverge analysis, we identified 7 genes under convergent acceleration in social branches (permulation $q$-value <0.15) (Fig. 3a, Supplementary Table 4) and 3 genes experiencing convergent deceleration in social branches (permulation $q$-value <0.15) (Fig. 3a, Supplementary Table 4). The 22 GO terms significantly associated with convergent acceleration (permulation $q$-value <0.15) were mainly related to neural function and programmed cell death (Fig. 3b), such as negative regulation of neuron death (GO:1901215), negative regulation of secretion (GO:0051048) and positive regulation of programmed cell death (GO:0043068) (Supplementary Table 5). The 7 GO terms significantly associated with convergent deceleration (permulation $q$-value < 0.15) were mainly associated with metabolism processes (Fig. 3c), such as aspartate family amino acid metabolic process (GO:0009066), sulfur amino acid metabolic process (GO:0000096) and methionine metabolic process (GO:0006555) (Supplementary Table 6).

We also used RELAX[43] to identify genes that experienced relaxed or intensified selection in all social branches relative to all nonsocial branches. We corrected the $p$-values reported by RELAX for multiple comparisons by computing $q$-values, as described above. Out of 7590 single-copy orthologs included in the RELAX analysis, more genes showed evidence of relaxation of selection (849 genes, K < 1, $q$-value <0.15, Fig. 3d, Supplementary Data 1) compared to intensification of selection (671 genes, K > 1, $q$-value <0.15, Fig. 3d, Supplementary Data 2). Genes experiencing significant relaxation of selection in social branches were enriched for 115 GO terms ($q$-value <0.15), including four main categories (Fig. 3e, Supplementary Data 3): DNA replication and repair (e.g., GO:0009451: RNA modification; GO:0008033: tRNA processing; GO:0006355: regulation of transcription, DNA-templated), reproduction (e.g., GO:0000003: reproduction, GO:0019953: sexual reproduction, GO:0032504: multicellular organism reproduction, GO:0007276: gamete generation), transcription (e.g., GO:0010467: gene expression; GO:0006396: RNA processing), development (developmental process: GO:0032502; multicellular organism development: GO:0007275), and metabolism (e.g., GO:0008152: metabolic process; GO:0043170: macromolecule metabolic process; GO:1901360: organic cyclic compound metabolic process). In addition, genes experiencing intensification of selection in social branches were enriched for 81 GO terms (Fig. 3f, Supplementary Data 4), including three main categories: immune response (e.g., GO:0032652: regulation of interleukin-1 production; GO:0002443: leukocyte mediated immunity), development (e.g., GO:0032502: developmental process; GO:0048856: anatomical structure development; GO:0009792: embryo development ending in birth or egg hatching; GO:0048729: tissue morphogenesis), and metabolism (e.g., GO:0071704: organic substance metabolic process; GO:0043170: macromolecule metabolic process; GO:0019538: protein metabolic process).

To clarify further what patterns of selection contribute to the acceleration and deceleration of protein evolution in social branches, we examined the overlap of lists of genes identified by RERconverge and RELAX. Two genes (*Protein vav-1, VAV1,* and *Protein SYS1 homolog, SYS1*) out of the seven genes experiencing significant convergent acceleration were also identified as experiencing significant relaxation, indicating that elevated rates of protein evolution for these genes in social branches is caused by relaxation of purifying selection.

### Social spiders harbor convergent amino acid substitutions

We finally asked if there were specific amino acid substitutions consistently associated with the convergent evolution of sociality in spiders. We used FADE[42] to identify sites experiencing directional selection towards specific amino acids in social compared to nonsocial branches[42]. After filtering sites that were also identified by FADE as

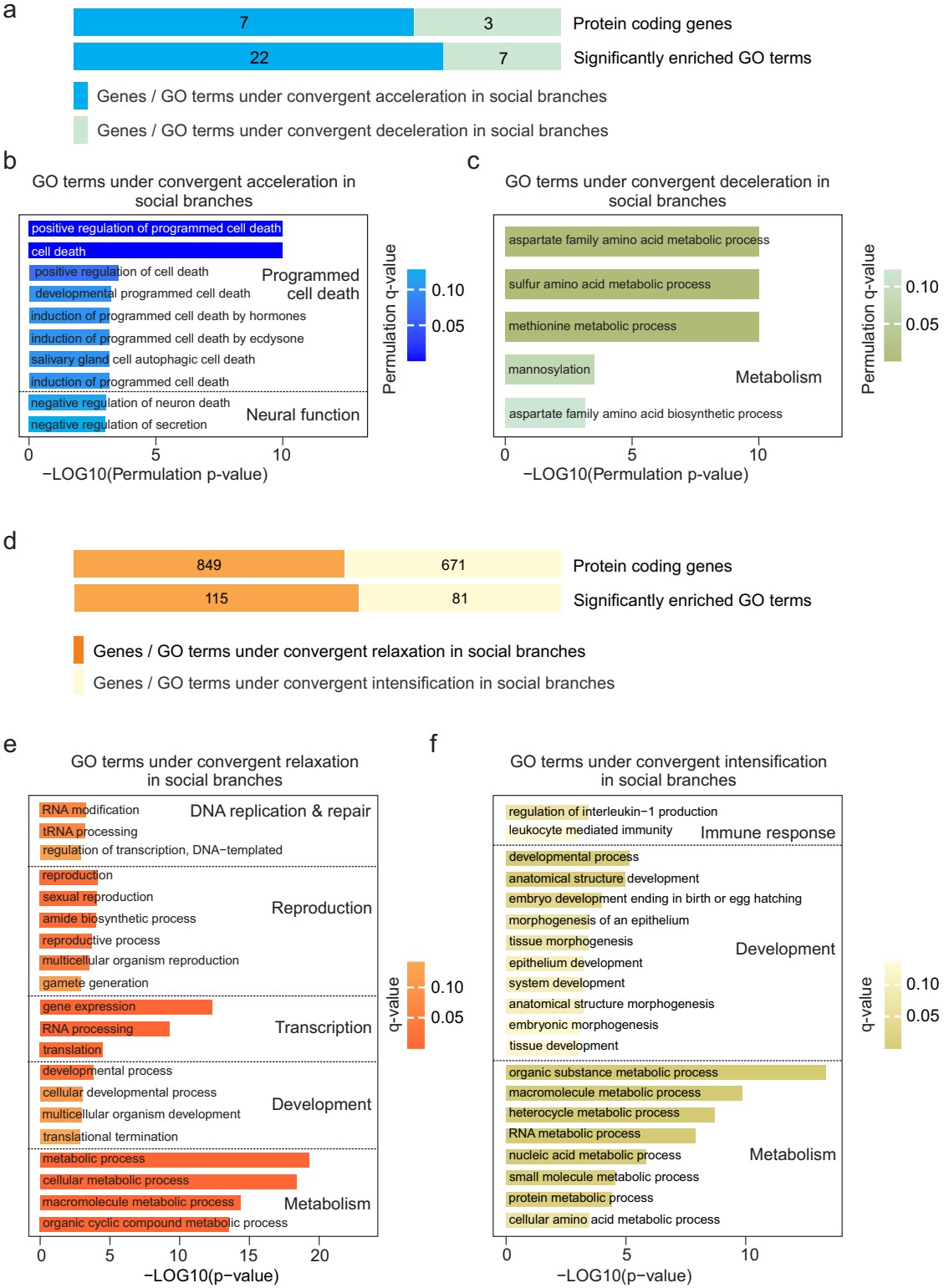

experiencing directional selection in nonsocial relative to social branches, we found 1421 sites in 123 genes with evidence (Bayes Factor > 100) for directional selection in social branches representing at least three independent origins of sociality. These genes were significantly enriched in 6 GO terms, such as positive regulation of neuromuscular junction development (GO:1904398), positive regulation of synaptic assembly at neuromuscular junction (GO:0045887), positive regulation of secretion (GO:0051047) and sterol homeostasis (GO:0055092) (Supplementary Table 7). Since we were most interested in identifying sites that showed a repeated pattern of substitution across the social branches, we further focused on sites that were inferred by FADE to have experienced the same substitution at least four separate times in social branches, resulting in a list of 357 sites in 27 genes (Supplementary Data 5). We further inspected protein alignments for these candidate genes and found that there were no substitutions that occurred in all social branches and no nonsocial branches. Substitutions that showed the strongest association with sociality (Supplementary Data 5) included substitutions in the

**Fig. 3 | Gene-level molecular evolution signatures of the convergent evolution of sociality in spiders. a** Genes that experienced convergent shifts in gene-wide rate of molecular evolution during the transition to sociality. The number of genes and significantly enriched GO terms (permulation *q*-value <0.15) experiencing convergent acceleration or deceleration in social branches identified by RERconverge. Genes or significantly enriched GO terms are depicted in light blue and light green, respectively. **b** GO terms with genes experiencing convergent accelerated evolution in social branches included categories such as neural function and programmed cell death. 10 out of 22 total significantly enriched GO terms are shown (Supplementary Table 5). **c** GO terms with genes experiencing convergent decelerated evolution concentratedly associated with metabolism functions. 5 out of 7 are shown (Supplementary Table 6). **d** A substantial set of genes that experienced convergent relaxation or intensification of selection during the transition to sociality. The number of genes and significantly enriched GO terms (*q*-value <0.15)

experiencing convergent relaxation or intensification of selection identified by RELAX. **e** Enriched GO terms for genes experiencing convergent relaxation of selection were associated with DNA replication and repair, reproduction, transcription, development, and metabolism functions (*q*-value <0.15). *P* value assigned to each enriched GO term was calculated by using Fisher's exact test and shown in Supplementary Table 5, the corresponding *q* value was calculated with Benjamini-Hochberg multiple test correction and shown in Supplementary Table 5. 20 out of 115 significantly enriched GO terms are shown. **f** Enriched GO terms for genes experiencing convergent intensification of selection were associated with immune response, development, and metabolism functions (*q*-value <0.15). *P* value assigned to each enriched GO term was calculated by using Fisher's exact test and shown in Supplementary Table 5, the corresponding *q* value was calculated with Benjamini-Hochberg multiple test correction and shown in Supplementary Table 6.

gene Bromodomain-containing protein 4 (Fig. S3) that occured in five social species, but also in the prolonged subsocial *D. cancerides* and the subsocial *A. arizona*. This gene also experienced other substitutions associated with social branches, providing some evidence for convergent evolution at the site-level in social spiders.

## Discussion

We used a comparative genomic approach with 22 spider species representing eight independent and recent origins of sociality to answer a longstanding question: what genomic changes underpin the convergent evolution of sociality? Overall, we identified genome-wide, genic, and site-specific changes that repeatedly occurred during recent convergent transitions to sociality in spiders. Our study shows that while the precise genetic changes vary across independent origins of sociality, the repeated evolution of sociality in spiders predictably leaves genome-wide signatures and involves genes with conserved functions that may often be involved in the evolution of social behavior.

Consistent with several previous studies in social spiders[23,26–29] and social insects[15,49,50], we found that social branches tend to experience elevated genome-wide rates of molecular evolution (i.e., dN/dS) compared to nonsocial branches. Such elevated genome-wide rates of molecular evolution are hypothesized to be caused in general by reduced effective population size in social species, which results in relaxation of purifying and positive selection experienced by genes[23,26–29]. In social spiders, a reduced effective population size should result from their inbred breeding system, reproductive skew, and female-biased sex ratios[22,23]. Indeed, we found that the genome-wide pattern was primarily driven by the relaxation of both purifying and positive selection (Fig. 2c), so that genes in social branches tend to experience more relaxed, neutral evolution when compared to genes in nonsocial branches. This genome-wide pattern is likely to be a longer-term consequence of the switch from outbreeding to inbreeding and from unbiased to female-biased sex ratios. There was also variation for genome-wide dN/dS that did not depend on sociality, which might be explained by lineage-specific differences in effective population size or other factors. Similarly, in a previous study in hymenopteran insects, social branches experienced elevated genome-wide dN/dS compared to nonsocial branches, but most variation existed between lineages, with bees—regardless social organization—having the highest dN/dS[50].

A fundamental phenotypic difference between social and non-social species is the extent of social interactions with conspecifics, which would have required increased behavioral tolerance towards conspecifics as an important preliminary change necessary for the evolution of cooperative group living in social spiders[25,51,52] and other animals[53,54]. Furthermore, shifts in behavioral development and the timing of social behaviors, reproductive behavior, and dispersal behaviors, are also hypothesized to be critical in the evolution of sociality in spiders[20,25] and insects[4]. Finally, changes in the size of

certain brain regions and sensory organs are hypothesized to be important for detecting and processing social signals and social information[55,56]. The shifts in rates of molecular evolution between social and nonsocial branches that we detected in genes associated with neural function, neurogenesis, behavior, and reproduction may underlie changes in these phenotypes. Several candidate genes stand out. For example, the gene *Autism susceptibility candidate 2* (*AUTS2*), which experienced intensified selection in social spiders, regulates neuronal and synaptic development and influences social behaviors in mice[56] and mutation in *AUTS2* has been associated with multiple neurological diseases, including autism in humans[57,58]. The gene *dystrophin, isoforms A/C/F/G/H* (*Dys*), which experienced intensified selection in social spiders, affects social behavior, communication, and synaptic plasticity in mice[59,60], and mutations in *Dys* are associated with autism and intellectual disability in humans[60]. The gene *Synaptogenesis protein syg-2* (*syg-2*), which experienced relaxed selection in social spiders, determines synapse formation, and mutations in *syg-2* are associated with locomotor behaviors in worms[61,62]. The genes *Syntaxin-1A* (*STX1*), *Syntaxin-5* (*STX5*), and *Syntaxin-6* (*STX6*), which experienced relaxed selection during social transition in social spiders, regulates the secretion of neurotransmitters and neuromodulators[63]. Each of these has been implicated in the expression of social behavior in various animals[64,65], and also implicated in abnormal behavioral phenotypes in humans, including autism and schizophrenia[66,67]. The gene VAV1, which experienced acceleration and also relaxation in social spiders, regulates rhythmic behavior in worms[68]. Further studies will be necessary to validate the function and mechanism of these candidates influencing spider social behavior.

Besides critical changes in neural and behavioral phenotypes, transitions to social living are also thought to be associated with phenotypic changes in disease pressure, hormone regulation, reproduction, and metabolism. Social living is hypothesized to be associated with increased disease pressure, although researchers have found mixed support for genetic changes in immune-related genes in social insects compared to solitary insects[69–71]. A set of genes involved in innate immunity, including *Pro-interleukin-16* (*IL16*) and *Toll-like receptor Tollo* (*TOLL8*) experienced relaxed selection in social spiders relative to nonsocial spiders. In addition, the gene *component C3* (*C3*), a key gene in the complement system of invertebrates, including spiders[72], experienced intensified selection in social spiders. These results are consistent with social spiders experiencing shifts in disease pressure, expected to be especially relevant given their inbred social system. Like social insects, the transition to social living in spiders has also been hypothesized to be associated with shifts in hormone regulation[73]. Genes experiencing relaxed selection in social spiders were enriched for GO terms including hormone–mediated signaling pathway and cellular response to steroid hormone stimulus, which could underlie these phenotypic changes. As emphasized above in terms of the genome-wide results, another phenotypic consequence of transitions to sociality in spiders, which may occur over longer time

scales, is the change of breeding system from outbreeding to inbreeding and the evolution of female-biased sex ratios[20,25]. Several candidate genes may be associated with these shifts in breeding system. For example, the gene *lilipod* (*lili*), which experienced intensified selection in social spiders, promotes self-renewal of germline stem cells during oogenesis in fly[74]. The gene *transcriptional regulator ovo* (*ovo*), which also experienced intensification in social spiders, regulates female germline development in mouse and fly[75–77]. Moreover, genes under relaxation of selection were significantly enriched for GO terms associated with reproduction and development, which could be tightly associated with the shifts in breeding system and sociality in social spiders. Metabolism-related genes and gene functions (GO terms) have also frequently been implicated in the expression of various types of social behavior in many animals[9,78], and comparative genomic studies in social insects have identified metabolism-associated genes as experiencing accelerated evolution in eusocial bees[79] and ants[80]. Consistent with these previous studies, we found that genes experiencing convergent shifts in selection in spiders were also enriched for metabolism-related functions.

In addition to gene-wide convergent signatures, although it is notoriously difficult to search for convergent amino acid substitutions, especially across a large number of genomes, we also identified some signatures of site-specific convergent molecular evolution. Genes with such convergent signatures of specific substitutions were enriched for several key GO terms, which were consistent with the functional categories that we found enriched at the gene level. However, inspection of the amino acid alignments for these genes showed that amino acid substitutions that occurred repeatedly in social branches also tended to be found in one or more nonsocial branches. For example, the gene *Bromodomain-containing protein 4* (*Brd4*), which has a set of specific convergent substitutions in at least five social spider species, affects regulating of inflammatory responses and learning and memory in rats[81]. The inhibition of *Brd4* can alleviate transcriptional dysfunction and Fragile X syndrome, a neurodevelopmental disorder that causes intellectual disability, behavioral deficits, and is a leading genetic cause of autism spectrum disorder in humans[82].

As described above, we identified genes and specific sites that tended to experience different patterns of molecular evolution in the replicate social branches compared to background nonsocial branches, with these genes being enriched for certain functions. However, we did not identify genes or sites within genes that *always* experienced evolutionary shifts in each of the eight branches that independently evolved sociality. For example, inspection of patterns of genic relative evolutionary rates (e.g., Fig. 3b, c) shows that genes with a relatively strong association between sociality and evolutionary rate still showed a lot of variation within each category. This means that we identified certain genes and functions that tended to experience shifts in evolutionary rates, but the exact details differed somewhat between the eight independent origins of sociality. Thus, specific substitutions or changes in specific genes are not required for the evolution of spider sociality, but changes do tend to occur in certain sets of genes or molecular pathways. Put another way, the strong statistical signatures across eight independent origins of sociality indicate that social spiders tend to travel similar evolutionary paths in terms of the functional genetic changes and genome-wide consequences, but the precise genetic details do vary. In practice, these results also emphasize that for complex phenotypes such as sociality, datasets representing many independent origins may be necessary to have the statistical power to detect these signatures. When samples representing multiple independent origins are not available, lineage-specific processes such as lineage-specific adaptations or historical contingency can be confounded with genetic changes that are actually associated with the phenotype of interest[16,30,83,84].

Our study adds to the growing list of studies that use comparative genomic approaches to elucidate the molecular underpinnings of convergent evolution and the degree to which similar phenotypic outcomes result from the same or similar genetic changes[46,85–88]. Collectively, these studies show that some convergent phenotypes, especially those involving relatively simple physiological adaptation, such as the evolution of antibiotic resistance, pesticide resistance, or resistance to neurotoxins[89], often predictably involve the same substitutions or changes in one family of genes. Other phenotypic changes that are relatively more complex, such as the evolution of sociality, still have identifiable genomic signatures across independent origins, but these signatures are not as highly predictable in the sense that they always involve the same substitution or changes in the same genes, but instead involve changes in genes with similar functions.

Finally, our study illustrates the potential for ethanol-preserved specimens to be used in transcriptome-based comparative genomics analysis. Increasingly, ethanol-preserved specimens are being used for DNA-based phylogenomic and comparative genomic studies[90], but RNA-based studies have usually been restricted to samples collected in liquid nitrogen or RNA later, and stored at −80 °C. Our study builds on previous studies that successfully extracted RNA from ethanol-preserved samples and subsequently conducted transcriptome-based phylogenomics[91,92]. We extracted RNA from spider specimens that had been preserved in ethanol at room temperature for years (Supplementary Table 1) and generated relatively high-quality RNA sequencing data (Supplementary Table 2). Collectively, our study highlights the heretofore underappreciated potential of RNA sequencing using ethanol-preserved specimens for comparative genomic and phylogenomic analyses, especially in species with large genomes, such as spiders[38], where sequencing and assembling whole genomes remains challenging.

## Methods
### Sample collection
We collected spider specimens of 16 species that range in social organization, including social, prolonged subsocial, subsocial and solitary, and stored the samples in 95% ethanol (Decon Labs, USA) (Fig. 1a). Specifically, this set of spider specimens includes five social and three subsocial species from the genus *Anelosimus* (Theridiidae), one social and four solitary species from the genus *Theridion* (Theridiidae), and one prolonged subsocial and two solitary species from the Sparassidae (Supplementary Table 1).

### RNA extraction with ethanol-preserved spider specimens
First, we selected one individual for each spider species, and dried the spider specimen using filter paper (Fig. 1a). Second, we transferred the dried specimen into a new 1.5 ml microcentrifuge tube (Thermo Fisher Scientific, USA), and grinded the sample into a fine powder using RNase-Free Disposable Pellet Pestles (Thermo Fisher Scientific, USA) on liquid nitrogen. Third, we added 900 μl Trizol (Invitrogen, USA) and 8 μl glycogen (Thermo Fisher Scientific, USA) to maximize the amount of pure RNA (Fig. 1a). Finally, we extracted total RNA from all selected individuals of 16 spider species following the Trizol manufacturer's protocol (Fig. 1a).

### Library preparation and RNA sequencing
We assessed the RNA quality with Agilent Bioanalyzer 2100 (Agilent Technologies, USA) using an RNA 6000 Pico Kit (Fisher Scientific, USA), and detected RNA quality with Nanodrop 1000 (NanoDrop Technologies, USA). Only RNAs with high quality were used for cDNA synthesis and amplification following the protocol as previously described[93]. Then, the primitive sequencing libraries were prepared and individually barcoded using the Nextera XT DNA Library Prep Kit (Illumina, USA) following the manufacturer's protocol. We further did size selection for the constructed libraries with approximately 550 bp inserted fragments. At the first step of size selection, 100 μl of 12% PEG-6000/NaCl/Tris and 10 μl prepared Dynabeads were added to the 20 μl

primitive library mixture and resuspended. The mixture was incubated for 5 min and placed on a magnetic stand for 5 min. The supernatant (150 μl) was transferred to a new tube and the beads discarded. At the second step to select fragments between 500 and 700 bp, 100 μl of 12.5% PEG-6000/NaCl/Tris and 10 μl prepared Dynabeads were added to the supernatant and mixed (Fig. 1a). The mixture was incubated for 5 min followed by bead separation on a magnetic stand. This time, the supernatant was discarded, and the beads were collected. The beads were washed twice with 70% ethanol (with 10 mM Tris, pH 6) and dried for 5 min. The tubes were then taken off the magnetic stand, and DNA was eluted from the beads by resuspending them in 15 μl EB. Finally, we measured the concentration of libraries using Qubit (Invitrogen, USA), and assessed the length distribution of libraries with Agilent Bioanalyzer 2100 (Agilent Technologies, USA). All libraries were sequenced on a single lane of an Illumina HiSeq 2500 platform (rapid run mode) in the DNA Sequencing Section at the Okinawa Institute of Science and Technology Graduate University using HiSeq Rapid SBS Kit v2 kit (500 cycles, Illumina, USA) yielding 250-bp paired-end reads (Fig. 1a).

### Additional omics data collection

Besides the new transcriptomic data from our 16 spider species, we downloaded available genome or transcriptome data of additional six closely related spider species from NCBI (Fig. 1b, Supplementary Table 2). Specifically, we included two species with whole genomes and another two species with transcriptomes from genus *Stegodyphus*, one species with transcriptomes from genus *Theridion* species and one species with transcriptome from family Sparassidae, *Heteropoda* spp. ATS12 (Supplementary Table 2).

### Assembly and annotation

We used FastQC (www.bioinformatics.babraham.ac.uk/projects/fastqc/) to assess sequencing read quality. We trimmed adapters and low-quality ends using Trimmomatic v.0.39 (github.com/timflutre/trimmomatic) until all reads scored above 20 at each position. We performed de novo transcriptome assembly using Trinity v.2.6.5[94] or rnaSPAdes[95] with default parameters, and removed the redundant transcripts in the assemblies for each species using CD-HIT v.4.8.1[96] with the threshold of 0.9, and further extracted the longest transcripts. Moreover, we performed the genome reassembly of *Anelosimus studiosus* using SPAdes[95], and further improved scaffolds of the draft genome using Redundans (github.com/lpryszcz/redundans), L_RNA_scaffolder[97] and PEP_scaffolder[98] with our newly sequenced *A. studiosus* transcriptome sequencing reads and annotated spider protein datasets following the analysis pipeline shown in Fig. 1b. Finally, we used BUSCO v.5.1.2 to assess the genome or transcriptome assembly completeness based on the arachnid_odb9 single-copy orthologous gene set from OrthoDB (www.orthodb.org) for each spider species (Fig. 1b).

We performed gene annotation by combining homology-based and de novo prediction approaches. First, we utilized AUGUSTUS[99] with spider *Parasteatoda tepidariorum* (GCF_000365465.3) as the training set, for de novo annotation (Fig. 1b). As for homolog-based prediction, we download protein sequences of published spider genomes from the NCBI database (Supplementary Table 2). The candidate genes were first identified by aligning these protein sequences to assembled transcriptomes and genomes using BLAT (github.com/djhshih/blat). We performed MAKER pipeline[100] to annotate the genome and transcriptome assemblies (Fig. 1b). Finally, we integrated the gene models predicted by both approaches using GLEAN[101] with default parameters to remove redundant genes.

### Ortholog identification

To assign the orthology among genes in above spider taxa, we used a best reciprocal hit search strategy to infer orthologous groups (OGs)[30,102]. In brief, we included the predicted complete proteomes

from the latest spider genomes (Supplementary Table 2). We pooled them into a local protein database and conducted a self-to-self BLAST search using DIAMOND v.0.9.29[103] with an E-value cutoff of 1e⁻⁵, and removed hits with identity <30% and coverage <30%. We identified the OGs from the BLAST results using OMA (www.omabrowser.org) with default settings, and finally inferred 3276 single-copy OGs. In addition, we downloaded the curated orthology map of Arachnida from OrthoDB (www.orthodb.org) which contains recorded 8805 OGs. Of these OGs in HaMStR v.1.0[104], we identified the putative OGs in each spider species with threshold E-values of less than 10⁻²⁰. We repeated this analysis pipeline twice, once with the 3276 OGs determined with OMA, and once with the 8805 OGs from OrthoDB. Among these putative OGs, we further identified one-to-one, one-to-many, and many-to-many orthologs among these spider taxa (Fig. 1b). For each 1:1 orthologous pair, we selected the longest transcripts associated with recorded OGs for each species as putative single-copy orthologs. Gene ontology (GO) terms were assigned to single-copy orthologs using eggNOG-mapper v2 (eggnog-mapper.embl.de/).

### Genome-scale phylogeny construction and divergence time estimation

To estimate the topology of the phylogeny including the 22 spider species and an outgroup *A. geniculata*, we used a phylotranscriptomic approach based on a single-copy ortholog dataset[105] (Fig. 1b). In brief, we prepared a dataset including all amino acid sequences of single-copy orthologs shared by the 23 spider species. Then, we performed sequence alignment using clustalo[106] and trimmed gaps using trimAl v1.2[107]. Moreover, we filtered each single-copy ortholog with strict constraints, including length (minimum 200 aa), sequence alignment (maximum missing data 50% in alignments). We prepared a concatenated dataset including core-shared single-copy orthologs and detected the best-fit model of sequence evolution using ModelFinder[108], and built the maximum likelihood phylogenetic trees using RAxML v8.2[109]. Statistical support for major nodes were estimated from 1000 bootstrap replicates. Finally, we estimated the divergence time using MCMCtree in PAML[41] with the topology of the 4DTV position and four calibration time based on the Timetree database (www.timetree.org) and published literatures[32].

### Analysis of codon substitution rate associated with sociality

To test whether the rate of molecular evolution was associated with sociality, we prepared the codon alignments of single-copy orthologs shared by 22 spider species, derived from amino acid sequence alignments and the corresponding DNA sequences using PAL2NAL v.14 (www.bork.embl.de/pal2nal/). We constructed a "supergene" dataset that used the concatenated codon sequences of all core-shared orthologs of 22 species, and another coalescent gene dataset that included all shared orthologs. We used PAML 4.7a[41] and HyPhy 2.5[42] to estimate the codon substitution rates across the spider phylogeny. First, we applied the free-ratio model ("several ω ratio") to calculate the ratio of non-synonymous to synonymous rate (dN/dS, ω) separately for each species with the supergene dataset using the package CodeML in PAML 4.7a[41] (Fig. 1b). To further characterize the patterns of molecular evolution associated with social organization (Fig. 1b), we estimated two discrete categories of dN/dS for social and nonsocial spiders taxa with the concatenated ortholog dataset (genome-wide) using HyPhy 2.5[42] (Fig. 1b). Finally, we employed a likelihood ratio test for the comparison of genome-wide dN/dS.

### Analysis of intensification or relaxation of selection associated with sociality

To elucidate the pattern of selection acting during the evolutionary transition to sociality, we used RELAX[43], which estimates variable dN/dS ratios across sites with three discrete categories. We compared

social spiders with nonsocial spiders at both genome-wide and gene-wide scales, and then employed a likelihood ratio test (LRT) (Fig. 1b). We corrected the *p*-values reported by RELAX for multiple comparisons by estimating *q*-values as an estimate of the false discovery rate (FDR) for each gene[48]. Genes showing K-value less than 1 and *q*-value <0.15 are considered to experience significant relaxation of selection. Similarly, genes showing K-values more than 1 and *q*-value <0.15 are considered to experience significant intensification of selection. Finally, we performed gene ontology (GO) enrichment analysis for these genes experiencing significant relaxation or intensification using GOATOOLS[110], and we corrected the reported *p*-values for multiple comparisons by computing *q*-values, and considered those GO terms (Biological Processes, BP) < 0.15 to be shown significant enrichment.

### Analysis of convergence shifts in relative evolutionary rates associated with sociality

To determine if particular orthologs experience convergent shifts in selective pressure, including acceleration or deceleration in social branches across the phylogeny, we estimated the protein evolutionary rate (relative evolutionary rate, RER) using the R package RERconverge[44] (Fig. 1b). RERconverge calculates relative branch lengths by normalizing branches for focal branches (i.e., social branches) to the distribution of branch lengths across all genes. This enables the identification of convergent changes in evolutionary rates across foreground relative to background branches while accounting for differences in phylogenetic divergence and in baseline rates of evolution across taxa. RERconverge compares rates of change in focal foreground branches and the rest of the tree and identifies genes that have a significant correlation between RERs and a phenotype of interest (e.g., sociality), as previously described[111,112]. In brief, we prepared the amino acid alignments of 7590 single-copy orthologs shared by up to 23 spider species. We estimated the branch lengths of each gene tree based on the inferred species tree using the R package phangorn[113], and calculated the RERs for each branch with the corresponding gene tree. We used 10,000 permulations to estimate permulation *p*-values for the correlation between the RER of foreground (i.e., social species) relative to the rest of the tree, and we corrected these permulation *p*-values for multiple comparisons by computing *q*-values[48]. We defined genes with significantly higher RERs (permulation *q*-value <0.15) in foreground branches as experiencing convergent acceleration in social branches, and genes with significantly lower RERs in foreground branches as experiencing convergent deceleration in social branches. In addition, we performed gene set enrichment analysis for the gene lists produced from the correlation analysis using the fastwilcoxGMTall functions in RERconverge with 10,000 permulations[44]. We corrected the resulting permulation *p*-values for multiple comparisons by computing *q*-values, and considered *q*-values <0.15 (i.e., GO terms with an FDR < 0.15) as significant.

### Analysis of convergent amino acid substitutions associated with sociality

To determine if convergent amino acid substitutions at specific sites of genes in social spider branches compared to nonsocial spiders across the phylogeny, we identified convergent substitution sites using FADE (FUBAR Approach to Directional Evolution) in HyPhy 2.5[42] with core-shared single-copy orthologs dataset (Fig. 1b). FADE identifies sites experiencing directional selection towards specific amino acids in foreground (i.e., social) relative to background (i.e., nonsocial) branches. We performed GO enrichment for genes with convergent sites using GOATOOLS as described above.

### Reporting summary

Further information on research design is available in the Nature Portfolio Reporting Summary linked to this article.

## Data availability

Raw and processed transcriptome data have been deposited in NCBI under the project PRJNA685164. We also used available spider genome and transcriptome data which were downloaded from NCBI, including *Acanthoscurria geniculata* (GCA_000661875.l), *Anelosimus studiosus* (GCA_008297655.l), *Stegodyphus mimosarum* (GCA_000611955.2), *Stegodyphus dumicola* (GCA_010614865.l), *Stegodyphus africanus* (SRR7062696), *Stegodyphus lineatus* (SRR7062695), *Theridion grallator* (SRR960715, SRR960716, SRR960718, SRR960719, SRR960611, SRR960612, SRR960614, SRR960615, SRR960616), *Heteropoda* sp. ATS12 (SRR6425926).

## Code availability

All scripts required to perform all analyses are publicly available on Github at github.com/jiyideanjiao/Social_Spider_Evolutionary_Genomics and Zenodo at https://doi.org/10.5281/zenodo.7222296[114].

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

## Acknowledgements
We would like to thank Dr. Marc Milne from University of Indianapolis for kindly contributing ethanol-preserved spider specimens. We would like to thank Dr. Yi-yong Zhao from Fudan University for suggestion in phylotranscriptomic analysis and Drs. Endo Tatsuya and Lijun Qiu from Okinawa Institute of Science and Technology Graduate University for expertise in RNA-seq library construction. This work was funded by the National Institutes of Health Grant GM115509 to T.A.L.

## Author contributions
C.T. and T.A.L. conceived and designed the study. L.A. and L.S.R. provided spider samples and expertise about spider natural history and social evolution. A.S.M. provided reagents and sequencing support. C.T. generated libraries for RNA sequencing. C.T. and T.A.L. performed all analyses and created all figures. C.T. and T.A.L. wrote a first draft of the manuscript and all authors contributed to the final manuscript.

## Competing interests
The authors declare no competing interests.
