## [Peer Review File · Nature Communications]

Genomic signatures of recent convergent transitions to social life in spidersREVIEWER COMMENTS

Reviewer #1 (Remarks to the Author):

This article examines the convergent genetic patterns associated with multiple recent transitions from a solitary to a social lifestyle in spiders. As the authors explain, this system is well-suited for this type of study, since sociality has evolved in parallel in multiple species across the spider phylogeny. However, one downside of the social spider system, which was acknowledged in the latter part of the discussion, is that sociality seems to be tightly associated with a transition to inbreeding in spiders. This is mentioned but not emphasized, although it seems to be an important factor given the results; as the authors state, inbreeding could account for a lot of the relaxation of selection. Overall, there are a lot of interesting results presented here, but the clarity of the manuscript could improve a lot with a substantial organizational overhaul.

Major suggestions

The authors close the abstract with this statement: "Our study also highlights the heretofore unappreciated potential of transcriptomics using ethanol-preserved specimens for comparative genomics and phylotranscriptomics." I agree that this is a potentially exciting contribution, but the methods lack even basic information about the quality and completeness of the transcriptomes. BUSCO scores and # of protein coding genes are provided in the supplementary materials file (Table 3), but I'd like to see more information about the method and the variation in transcriptome quality (among samples and relative to transcriptomes prepared using standard methods), especially given the emphasis in the article. I also think it would be appropriate to report this information in the main text.

In general, the comparative analysis methods and variation in sample sizes were not explained in sufficient detail. For instance, the authors emphasize the single gene (TPC1) that has a convergent amino acid substitution in all social species. For other cases with some evidence of convergence, it would be useful to report what proportion of the genes in each category were determined to be convergent in two social lineages, three social lineages, etc. In the example shown in figure 2 (ITA2), for instance, the 'convergent acceleration' appears to be driven primarily by sister species (?) *A. oritoyacu* and *A. guacamayos*. Maybe I missed or misunderstood something, but I was not able to extract this information from the supplementary tables.

I don't have firsthand experience with some of these analyses (RELAX, RER), but is it feasible to address the likelihood of retaining false positives?

The organization of the discussion did not follow the same structure as that of the results. In my opinion, this leads to a strong emphasis on the more speculative results from the analyses, those focused on the long list of genes that may be associated with social transitions and the possible functions of those genes with respect to sociality. While the study highlights some potentially useful candidate genes*, the more concrete results indicate that relaxed selection contributes strongly to the genome-wide increased

dN/dS rate observed in the social spider lineages (figure 1d). In line with this, I think it's important to acknowledge earlier in the discussion the other characteristics exhibited by social species (inbreeding, female-biased sex ratio), as these likely play a large role in shaping the reported patterns.

* If I wanted to investigate these candidate genes in a focal species or two, I didn't easily see how I could access information about the strength of selection by species. Could this be added to the supplementary tables?

Figures 2d and 2e: The figure legends do not provide sufficient detail to permit complete interpretation of these figures. What are the unlabeled blue dots closest to the x-axis? In both examples, it looks like one or two species shape the reported patterns- can this be quantified?

Figures 3d and 3e: I struggled to understand these plots. I eventually found a clear description in a supplementary figure legend, but it would really help to have a clear explanation in the main article.

Minor issues

Line 103: I didn't see a *Stegodyphus* sample listed in Supplementary Table 1- is this statement correct or is there a sample missing?

Line 112: It wasn't fully clear how this statement aligns to the information provided in the cited supplementary table.

Line 163: The number of single copy orthologs listed here is not the same as the one reported on line 110. A different subset of single copy orthologs are used for the subsequent RELAX analysis as well, listed line 179. What were the criteria for removal in each case?

Reviewer #2 (Remarks to the Author):

The paper investigates molecular evolution across multiple origins of sociality in spiders, using comparative analysis of 24 species, using both transcriptome and genome sequencing. They provide evidence of elevated genome-wide rates of molecular evolution in social species vs non-social species. They also report an interesting example of convergent evolution of a specific substitutions in social vs non-social species, which is quite novel.

I found the paper to be well-written, and the analyses valid and described in sufficient detail.

Below, I provide comments for the authors to consider to clarify and strengthen their manuscript:

Is spider sociality a major transition in evolution? Since they are not eusocial, I am unclear on whether this can be argued to be a major transition, as written up front in the abstract. Please provide

justification for this.

The abstract refers to genomes, but the data come from both transcriptomes and genomes. I appreciate that the authors used ethanol-preserved specimens, but I am somewhat concerned this may lead to a “mixed bag” of coverage of the genome/transcriptome across species that could bias the results. Can the authors provide justification that they had fair representation of gene sets from all species/specimen preparation types, and if not, then how did they account for this in their analysis?

Using ethanol for RNA work is surprising. Can the authors provide some more information on RNA quality, and some additional metrics on how many samples/mass used to get the amount of RNA needed for size selection, etc? Could the size selection bias the data in any way against certain parts of the genome or genes? Wondering if these transcriptomes are “representative” of the whole transcriptome or not.

The authors describe variation in level of sociality of the spiders; but is this accounted for in any way in the analysis? Is it possible to test for further breakdown of patterns by type of sociality? I think this may be informative since different social forms may experience different demographic shifts and ecological drivers, associated with different patterns of molecular evolution.

What is the divergence time of these spiders? This information should be provided up front in the main text.

The authors use RELAX to discriminate genes with relaxed selection vs intensified, which is a nice approach. However, they do not carry through with their interpretation of these different classes of genes (e.g. paragraph beginning with line 261). Can they provide more of a distinction between genes with relaxed vs accelerated selection? They are all discussed together without distinction to their unique patterns of evolution, which muddles the discussion about the evolutionary trajectory of each of the genes.

Reviewer #3 (Remarks to the Author):

In their study, the authors sequenced and assembled many new transcriptomes from social spider species with the goal of understanding which genetic changes are associated with the switch from solitary to social life. They compared those social, semi-social, and solitary species across a phylogeny to identify genome-wide patterns of differential selection, gene-specific patterns of accelerated selection, and convergent amino acid substitutions associated with social life. They identified a global pattern of relaxed selection in social species. They also report a general acceleration due to relaxation of constraint on genes in pathways related to neurogenesis and behavior.

Generally, I find the paper to address an interesting subject that readers will appreciate. The approaches

here had not previously been applied to social life, to my knowledge. Their contributions in just genomics is substantial, and their methods are generally sound. However, I present major concerns detailed below that need to be addressed such that all aspects are up to field standards in terms of statistics and presentation to the reader.

Major Concerns:

1. The inferred species tree would be best reported with branch support values in Fig S1 so that the reader can understand which clades are strongly supported. Possibly even in Figure 1, such as with dots on nodes over a particular support value. Branch support values could be bootstrap support or approximate likelihood ratio test (PhyML).
2. When reporting the enriched GO terms for the RERconverge and RELAX analyses, they should be filtered by and reported with their adjusted p-values or false discovery rates (FDRs). An FDR higher than 0.05 can be used. Such an adjustment would affect Figures 2 and 3, and some of the manuscript text. Judging by Supp Tables 5, 6, 8, and 9, there are still some relevant categories that pass adjustment, and using FDRs could help focus conclusions on the finding with the best statistical support.
3. Positive comment. The significant overlap between the RELAX and RERconverge results nicely bolsters the authors conclusions.
4. The TPC1 study. Could the authors comment on where TPC1 is expressed? Neuronal? Brain? Is it known to affect behavior or social behavior in arthropods?
5. Searching for convergent amino acid substitutions out of the whole genome is notoriously difficult. Does FADE provide a way of understanding how many apparently convergent changes would be expected by chance? Did the authors see a relative excess compared to that expectation?

Author Rebuttals to Initial Comments:

Thank you very much for the opportunity to revise and resubmit our manuscript NCOMMS-21-10286-T to *Nature Communications*. We thank the three Reviewers and the Associate Editor for their constructive comments and have carefully considered each of their comments.

In line with Reviewer suggestions, we have completed an extensive reanalysis and revision of the original manuscript. Specifically, in the revision, we reassembled all of the spider transcriptomes using integrative approaches (Trinity and rna-SPAdes) and improved the draft genome assembly of the *Anelosimus studiosus* genome (GCA_008297655.1). We also decided to remove the transcriptomes of two spider species (*Theridion californicum* and *Caayguara ybytyriguara*) from our study because these two published transcriptomes had a very high rate of fragmentation for single-copy genes (> 30%). Removing these two relatively low-quality transcriptomes, together with our steps to improve the transcriptome and genome assemblies for the remaining species significantly improved our dataset and downstream analyses. For example, the transcriptomes and genomes in our revised study include an estimated average of 91.24% of highly conserved single-copy genes found across arachnids, demonstrating that our assemblies are relatively very complete and with high BUSCO scores. To further clarify the analysis pipeline that we used, we also added a flowchart to the revised manuscript as **Fig. 1b**. The main results of our revised analyses remain the same as reported in the original manuscript, but some of the specifics (e.g., genes with signatures of convergent substitutions) have changed. We have also updated all the scripts we used for the analyses and posted them on GitHub (www.github.com/jiyideanjiao/social_spider_genome).

We believe that, as a result of these revisions, our manuscript is greatly improved. We have addressed all the comments and suggestions in the detailed point-by-point list below:

Responses to comments of Reviewer 1

R1-Q1: This article examines the convergent genetic patterns associated with multiple recent transitions from a solitary to a social lifestyle in spiders. As the authors explain, this system is well-suited for this type of study, since sociality has involved in parallel in multiple species across the spider phylogeny. However, one downside of the social spider system, which was acknowledged in the latter part of the discussion, is that sociality seems to be tightly associated

with a transition to inbreeding in spiders. This is mentioned but not emphasized, although it seems to be an important factor given the results; as the authors state, inbreeding could account for a lot of the relaxation of selection. Overall, there are a lot of interesting results presented here, but the clarity of the manuscript could improve a lot with a substantial organizational overhaul.

Response: Thank you very much for these positive comments. Sociality in spiders indeed is tightly associated with inbreeding. We do not see this as a “downside” of the social spider system, but simply a fact of nature. Notably, genome-wide relaxation of selection caused by reduced effective population size seems to very generally be associated with the evolution of sociality across animals. Put another way, relaxation of selection may be even more extreme in social spiders because they also inbreed (further reducing effective population size), but the association between sociality and reduced effective population size is definitely not unique to spiders. In the revised manuscript, we further emphasized effects of inbreeding and the association between sociality and inbreeding. In the revised Discussion, we wrote: “In social spiders, inbreeding in particular is expected to cause reduced effective population size, and female-biased sex ratios and reproductive skew are expected to further reduce effective population size. Indeed, we found that the genome-wide pattern was primarily driven by the relaxation of both purifying selection and positive selection (**Supplementary Fig. 2c**), so that genes in social branches tend to experience more relaxed, neutral evolution when compared to genes in nonsocial branches. This genome-wide pattern is likely to be a longer-term consequence of the switch from outbreeding to inbreeding and also from the switch from unbiased to female-biased sex ratios, which likely occurred after the origin of sociality.”

R1-Q2: The authors close the abstract with this statement: "Our study also highlights the heretofore unappreciated potential of transcriptomics using ethanol-preserved specimens for comparative genomics and phylotranscriptomics." I agree that this is a potentially exciting contribution, but the methods lack even basic information about the quality and completeness of the transcriptomes. BUSCO scores and # of protein coding genes are provided in the supplementary materials file (Table 3), but I'd like to see more information about the method and the variation in transcriptome quality (among samples and relative to transcriptomes prepared using standard methods), especially given the emphasis in the article. I also think it would be appropriate to report this information in the main text.

Response: We are very grateful that you appreciated the approach of RNA sequencing with ethanol-preserved specimens. We were initially surprised by how well this worked. The potential for RNA sequencing using ethanol-preserved specimens has been largely unexplored, and incidentally, the potential to use available ethanol-preserved samples became more important for us during the COVID-19 pandemic, when traveling to collect fresh material was not possible. As requested by the Reviewer, in the revised manuscript, we have added specific details regarding the RNA extraction and library construction protocols in the main text and in the Supplemental Materials. As shown in **Supplementary fig. 1**, RNA extracted from the ethanol-preserved specimens was still relatively complete, as detected by Agilent Bioanalyzer 2100. For RNA extraction, we followed the standard Trizol manufacturer's protocol, without any additional steps. For the RNA sequencing library construction, we prepared the libraries following the protocol of the Mikheyev lab at the Okinawa Institute of Science and Technology Graduate University, as previously described (Aird *et al.* 2013, *BMC Genomics*). As shown in **Supplementary fig. 2**, the resulting constructed RNA-seq libraries were of similar high quality as other libraries prepared by standard methods. We believe that other researchers will also be able to use this approach to extract relatively high quality RNA from spider specimens and other arthropod specimens. Indeed, we have subsequently used the same protocol to successfully extract and sequence RNA from ants preserved in ethanol.

We apologize for the unclear description of BUSCO scores, etc. in our initial manuscript. For the completeness of genome content, we used the BUSCO pipeline to assess the quality of transcriptome assembly. To clarify, we used BUSCO to assess the transcriptome assembly completeness based on the arachnid_odb9 dataset, including 2,934 single-copy orthologous genes. We found an average of 91.24% arachnid conserved genes in each spider species (**Supplementary Table 3**), demonstrating that the transcriptomes we generated from ethanol-preserved samples are relatively high quality. Further, we used an integrative approach combining OMA (<https://omabrowser.org/>) and OrthoDB (<https://www.orthodb.org/>) pipelines to identify a total of 7,590 orthologs across the 22 spider species in our study, and an average of 6,723 (88.58% of the total) orthologous genes in each new spider species (**Supplementary Table 3**). We believe that these results demonstrate that the transcriptome assemblies are of relatively high quality, and suitable for subsequent comparative genomic analyses that we performed, focused at the protein-coding gene level.

R1-Q3: In general, the comparative analysis methods and variation in sample sizes were not explained in sufficient detail. For instance, the authors emphasize the single gene (TPC1) that has a convergent amino acid substitution in all social species. For other cases with some evidence of convergence, it would be useful to report what proportion of the genes in each category were determined to be convergent in two social lineages, three social lineages, etc. In the example shown in figure 2 (ITA2), for instance, the 'convergent acceleration' appears to be driven primarily by sister species (?) *A. oritoyacu* and *A. quacamayos*. Maybe I missed or misunderstood something, but I was not able to extract this information from the supplementary tables.

Response: Thank you for pointing out these issues which caused confusion. To clarify the analysis pipeline that we used, we added a flowchart to the revised manuscript as **Fig. 1b**.

We conducted convergent shift tests at the genome-, gene- and site-wide levels. For the gene-wide RERconverge analysis (Kowalczyk *et al.*, 2019. *Bioinformatics*), we tested for signature of convergent molecular evolution underlying convergent phenotypic evolution, i.e., across multiple independent origins of sociality. RERconverge examines the rate of protein sequence evolution (relative evolutionary rate, RER) for each gene on every branch across the phylogeny, standardized to the distribution of rates across all genes. RERconverge seeks to identify genes that **on average** experience accelerated or decelerated rates of molecular evolution **across multiple independent origins** of a trait of interest. The power of the RERconverge approach is that it identifies statistical signatures of convergent molecular evolution across multiple lineages and multiple independent origins of the trait of interest. RERconverge does not separately identify genes that are associated with the origin of the trait of interest for each independent origin. Multiple origins are needed to have the statistical power to identify changes associated with the repeated origin of the trait, as opposed to other lineage-specific phenotypic changes: specifically, the RERconverge package requires a minimum of 5 species and 3 independent origins of the trait of interest, although of course more are preferable. Thus, it is not possible to use our dataset to quantify what proportion of genes are repeatedly found across independent origins of sociality, although we agree that this is a very interesting question. We would need a much larger dataset, e.g., with 5+ species and 3+ independent origins of sociality *for multiple spider lineages*, in order to be able to quantify the degree to which the same or similar genetic changes were repeatedly associated with the convergent evolution of sociality across spider lineages.

Please note that we tried to carefully explain these issues in the Discussion, by emphasizing that the RERconverge approach identifies statistical signatures of genes that on average experience accelerated or decelerated rates of molecular evolution across independent origins of sociality. Specifically, we wrote: “As described above, we identified genes and specific sites that tended to experience different patterns of molecular evolution in the replicate social branches compared to background nonsocial branches, with these genes being enriched for certain functions. However, we did not identify genes or sites within genes that *always* experienced evolutionary shifts in each of the eight branches that independently evolved sociality.”

For the gene-wide RELAX analysis (Wertheim *et al.*, 2014. *Molecular Biology and Evolution*), we tested for relaxed or intensified selective pressure acting on average on social branches relative to background (i.e., nonsocial) branches across the spider phylogeny. RELAX begins by fitting a codon model with three ω classes to the entire phylogeny (null model). RELAX then tests for relaxation or intensification by introducing the parameter k (where $k \geq 0$), serving as the selection intensity parameter, as an exponent for the inferred ω values, ω^k . Specifically, RELAX fixes the inferred ω values ($\omega_1, \omega_2, \omega_3$) and infers, for the foreground branches, a value for k which modifies the rates to $\omega^k <1,2,3>$ (alternative model). RELAX then conducts a Likelihood Ratio Test to compare the alternative and null models (<https://stevenweaver.github.io/hyphy-site/methods/selection-methods/>). A significant result of $k > 1$ indicates that selection strength has been intensified along the social branches, and a significant result of $k < 1$ indicates that selection strength has been relaxed along the social branches (**Additional figure 1**). As we introduced in the main text, relaxed selection (i.e., relaxation) is caused by weakening of both purifying selection ($\omega < 1$) and positive selection ($\omega > 1$), towards neutrality (see the arrow in **Additional figure 1**). Intensified selection (i.e., intensification) is caused by strengthening of both purifying selection ($\omega < 1$) and positive selection ($\omega > 1$), away from neutrality (see the arrow in **Additional figure 1**). The RELAX analysis is using the entire phylogeny of a specific gene, and considering all the social branches as a whole, therefore we defined the genes with significant K value ($P < 0.05$) experiencing convergent relaxation or intensification.

In the revised manuscript, we generated a new **Fig.3** to summarize the genes and enriched GO terms experiencing convergent shifts in selection, and updated the lists in **Supplementary Table 5, 6, 7, 8**. We think that this new figure helps to explain the RELAX approach.

Additional figure 1. Schematic diagrams depicting the cases of a genes under relaxation or intensification.

R1-Q4: I don't have firsthand experience with some of these analyses (RELAX, RER), but is it feasible to address the likelihood of retaining false positives?

Response: For the use of RELAX to test relaxed or intensified selection, we followed the author guidelines (<https://stevenweaver.github.io/hyphy-site/methods/selection-methods/>). The authors have shown that their approach is appropriate to deal with the issue of false positives (Wertheim et al., 2015, *Molecular Biology and Evolution*), and RELAX has been very widely used (e.g., Liang et al., 2021, *Current Biology*; Chark et al., 2021, *Molecular Biology and Evolution*; Sun et al., 2021, *Molecular Biology and Evolution*; Cui et al., 2021, *Molecular Ecology*; Cui et al., 2021, *Cell*).

For the use of RERconverge, we followed the author guidelines (<https://github.com/nclark-lab/RERconverge>). The authors of the RERconverge package have previously shown that their approach is appropriate to control the rate of false positives (Kowalczyk et al., 2020, *eLife*; Partha et al., 2019, *Molecular Biology and Evolution*; Kowalczyk et al., 2019, *Bioinformatics*). RERconverge has increasingly widely been used in recent studies (e.g., Kolora et al., 2021. *Science*; Daane et al., 2021. *Current Biology*; Zou et al., 2021. *Molecular Biology and Evolution*; Rubin et al., 2019. *Philosophical Transactions of the Royal Society B*). Specifically, we performed the permutation analysis that has recently been recommended by the authors of RERconverge to correctly control the rate of false positives and control for the observed

phylogenetic structure (11 January, 2021, <https://github.com/nclark-lab/RERconverge/blob/master/vignettes/PermutationWalkthrough.pdf>). We revised the main text of the manuscript to further explain the permutation approach. In our study, we defined the genes significantly experiencing convergent acceleration or deceleration based on the permutation P value < 0.05 , and not the standard nominal P value or adjusted P values. Please also see our response to **R3-Q2**, where we further explain the permutation approach and discuss the issue of false positives.

R1-Q5: The organization of the discussion did not follow the same structure as that of the results. In my opinion, this leads to a strong emphasis on the more speculative results from the analyses, those focused on the long list of genes that may be associated with social transitions and the possible functions of those genes with respect to sociality. While the study highlights some potentially useful candidate genes*, the more concrete results indicate that relaxed selection contributes strongly to the genome-wide increased dN/dS rate observed in the social spider lineages (figure 1d). In line with this, I think it's important to acknowledge earlier in the discussion the other characteristics exhibited by social species (inbreeding, female-biased sex ratio), as these likely play a large role in shaping the reported patterns.

* If I wanted to investigate these candidate genes in a focal species or two, I didn't easily see how I could access information about the strength of selection by species. Could this be added to the supplementary tables?

Response: We agree that the strongest pattern we found was the pattern of genome-wide relaxation of selection in social relative to nonsocial branches, consistent with many other studies. We also agree that ideally that order of presentation of the Results is the same as the Discussion. Thus, as suggested, in the revised manuscript Discussion, we have moved discussing the genome-wide results before discussing the gene-level results. However, we only agree that our gene-level results are “speculative” in the sense that we have to rely on inferred functional annotation from model systems. The RERconverge and RELAX pipelines we have used provide strong statistical evidence for genes showing shifts in rates of molecular evolution in social versus nonsocial branches. These pipelines are widely used. Our dataset is also much larger (i.e., including more total species, and representing more independent origins of our focal trait) than most other studies (e.g., the previous largest comparative genomic study of sociality, in bees, included 10 species representing 2 independent origins and 2 independent elaborations of eusociality; Kapheim et al. 2015, *Science*)

As requested, we added more information about the significant genes in supplementary tables, including the relative evolutionary rate values for each branch for each gene (**see Supplemental dataset1**).

R1-Q6: *Figures 2d and 2e: The figure legends do not provide sufficient detail to permit complete interpretation of these figures. What are the unlabeled blue dots closest to the x-axis? In both examples, it looks like one or two species shape the reported patterns- can this be quantified?*

Response: We agree that these plots (that are standard in publications using the RERconverge package) are not as clear as they should be. The unlabeled blue dots closest to the x-axis in our previous **Fig. 2** were the dots representing the estimated relative evolutionary rates (RERs) of ancestral nodes. To avoid causing further confusion for readers, we decided to remove these plots, and we provide more detailed information about estimated RERs for each gene as a new **Supplementary dataset1**. As we clarified earlier, we defined genes tended to have higher RERs in social branches than nonsocial branches as genes experiencing convergent acceleration, vice versa.

R1-Q7: *Figures 3d and 3e: I struggled to understand these plots. I eventually found a clear description in a supplementary figure legend, but it would really help to have a clear explanation in the main article.*

Response: We have moved the Venn diagram and explanation to the revised main text as a new **Fig. 3g**.

R1-Q8: *Line 103: I didn't see a Stegodyphus sample listed in Supplementary Table 1- is this statement correct or is there a sample missing?*

Response: We did not generate new RNA sequencing data of *Stegodyphus* samples. We used published transcriptomes of two *Stegodyphus* species, *S. africanus* and *S. lineatus* (Bechsgaard et al., 2019, *Molecular Biology and Evolution*), and published genomes of another two *Stegodyphus* species, *S. mimosarum* (Sanggaard et al., 2014, *Nature Communications*) and *S.*

dumicola (Liu et al., 2019, *Genes*). We have revised the manuscript and the Supplementary Table to clarify this.

R1-Q9: Line 112: It wasn't fully clear how this statement aligns to the information provided in the cited supplementary table.

Response: Thanks for pointing this out. We now revised this statement and updated the **Supplementary Table 3**.

R1-Q10: Line 163: The number of single copy orthologs listed here is not the same as the one reported on line 110. A different subset of single copy orthologs are used for the subsequent RELAX analysis as well, listed line 179. What were the criteria for removal in each case?

Response: In the revised manuscript, as described above, we significantly improved transcriptome assemblies and genome reassemblies for the included species. This allows us to identify more orthologs in each spider species. Thus we could use all 7,590 shared orthologs for both RERconverge and RELAX analyses. We also correct these issues in the revised manuscript.

Responses to comments of Reviewer 2

The paper investigates molecular evolution across multiple origins of sociality in spiders, using comparative analysis of 24 species, using both transcriptome and genome sequencing. They provide evidence of elevated genome-wide rates of molecular evolution in social species vs non-social species. They also report an interesting example of convergent evolution of a specific substitutions in social vs non-social species, which is quite novel.

I found the paper to be well-written, and the analyses valid and described in sufficient detail.

Below, I provide comments for the authors to consider to clarify and strengthen their manuscript:

Response: Thank you very much for your positive comments and suggestions. We believe that social spiders present an excellent system to study the convergent evolution of sociality.

R2-Q1: *Is spider sociality a major transition in evolution? Since they are not eusocial, I am unclear on whether this can be argued to be a major transition, as written up front in the abstract. Please provide justification for this.*

Response: There is a great deal of debate in the literature about exactly how to categorize social organization across animals (e.g., Rubenstein and Abbot 2017 argue that broader definitions are often beneficial for comparative analyses) and also what counts as a “major transition”. Maynard Smith and Szathmáry (1995), which originally laid out the major transitions also had issues and inconsistencies: for example, while each of the subsequent major transitions generally builds off of the previous, humans are usually not considered to be eusocial, but the evolution of language in human societies is labeled as a major transition. In any case, whether or not spider sociality counts as a “major transition” is not an important issue for our manuscript, and to avoid any confusion, we have rewritten the text. Specifically, in the Abstract, we revised the text to: “The transition from solitary to social life is a major phenotypic innovation, but its genetic underpinnings are largely unknown.” In the Introduction, we revised the text to: “The evolution of sociality is a phenotypic innovation that occurred repeatedly and sporadically across vertebrates, insects, spiders, and crustaceans (Rubenstein and Abbot 2017)”

R2-Q2: *The abstract refers to genomes, but the data come from both transcriptomes and genomes. I appreciate that the authors used ethanol-preserved specimens, but I am somewhat concerned this may lead to a “mixed bag” of coverage of the genome/transcriptome across species that could bias the results. Can the authors provide justification that they had fair representation of gene sets from all species/specimen preparation types, and if not, then how did they account for this in their analysis?*

Response: This is an excellent point. The quality of transcriptome or genome assembly is very important for comparative genomic analysis. In our previous manuscript, we did include two relatively low quality published transcriptome assemblies of *Theridion californicum* and *Caayguara ybytyriguara*, showing a high fragmented proportion of single-copy genes (> 30%). In the revised manuscript, we excluded these two relatively low quality transcriptomes. In the past months, we also sought to improve the quality of the remaining transcriptome assemblies. For this, we used another transcriptome assembly toolkit, rna-SPAdes (<https://github.com/ablab/spades>) to assemble all the spider transcriptomes, and finally obtained

better *de novo* assemblies than our previous version (**Supplementary Table 3**). In addition, we reassembled the genome of *Anelosimus studiosus* (GCA_008297655.1) and further improved the scaffolds. As shown in **Supplementary Table 3**, we detected an average of 91.24% arachnid conserved genes (2,934 genes) in each spider species, ranged from 84.60% to 99.0%. Further, we identified an average of 6,854 orthologs in each spider species, ranging from 6,165 to 7,590 orthologs. This is much better than a recent comparative genomics study in spiders (Schwager *et al.*, 2017. *BMC Biology*), which identified a maximum of 3,592 shared orthologs in spider genomes (**Additional file 1: Table S1**). Although most data in our study are from transcriptomes, our main analyses are based on protein-coding genes. These analyses work very well using high-quality transcriptome assemblies.

R2-Q3: Using ethanol for RNA work is surprising. Can the authors provide some more information on RNA quality, and some additional metrics on how many samples/mass used to get the amount of RNA needed for size selection, etc? Could the size selection bias the data in any way against certain parts of the genome or genes? Wondering if these transcriptomes are “representative” of the whole transcriptome or not.

Response: We were also initially surprised by how well this worked. The potential for RNA sequencing using ethanol-preserved specimens has been largely unexplored, and incidentally, the potential to use available ethanol-preserved samples became more important for us during the COVID-19 pandemic, when traveling to collect fresh material was not possible. In detail, we extracted RNA from one individual spider for each species and we followed the Trizol manufacturer's protocol. Although the spiders we included have relatively small body sizes, we were able to obtain enough RNA (> 2,000 ng, 30 µl) for sequencing library construction. We constructed the RNA-seq library following the protocol of Mikheyev lab at the Okinawa Institute of Science and Technology Graduate University as previously described (Aird *et al.* 2013, *BMC Genomics*), and successfully produced high-quality RNA-seq libraries for each spider species (see **Supplementary Fig. 2**). Size selection is a very standard step in RNA-seq library preparation and a library usually includes a wide range of inserted fragments, for example, from 100 nt to 1,200 nt. For most Illumina sequencers, the instrument can only read less than 300 nt inserted fragment at both directions, called 150-bp paired end (such as Novaseq 6000), 250-bp paired end (such as HiSeq 2500, rapid run mode), 300-bp paired end (such as MiSeq). That is, very long (instrument only reads the header and the tail of the fragments) or very short inserted fragments will use a large fraction of output reads, while not significantly helping the

transcriptome assembly. In this study, we originally planned to use a MiSeq sequencer (300 paired end) but the maximum data output is 15 Gb. To run all of the libraries on a single lane of the same sequencer, we used an Illumina HiSeq 2500 platform (rapid run mode) with a HiSeq Rapid SBS Kit v2 kit (500 cycles) yielding 250-bp paired-end reads. We used a size selection step to select the inserted fragments between 500 and 700 bp, and took full advantage of the sequencer outputting 250-bp paired-end reads. Finally, we used integrative approaches (Trinity and rna-SPAdes) to de novo assemble the transcriptome and obtained relatively high quality of assembly (BUSCO detected average of 91.24% arachnid conserved single-copy genes), indicating these processes to be effective. These results, and the fact that we identified an average of 6,854 orthologs in each spider species demonstrate that the transcriptomes we assembled are “representative” of the whole transcriptome.

R2-Q4: The authors describe variation in level of sociality of the spiders; but is this accounted for in any way in the analysis? Is it possible to test for further breakdown of patterns by type of sociality? I think this may be informative since different social forms may experience different demographic shifts and ecological drivers, associated with different patterns of molecular evolution.

Response: We agree that this would be very interesting, but unfortunately it is not possible with our current dataset. Specifically, our current dataset is excellent for comparing social versus nonsocial (including an estimated 8 independent origins of sociality), as defined in the manuscript, but there are not enough independent origins of subsociality (=two) or prolonged subsociality (=one) in our dataset to have the statistical power to identify signatures of convergent molecular evolution associated with the evolution of subsociality or prolonged subsociality. In a future study, we hope to include a large number of samples, in particular with more representatives of subsocial and prolonged subsocial species, to further tease apart how shifts in levels of spider sociality are associated with shifts in rates of molecular evolution.

R2-Q5: What is the divergence time of these spiders? This information should be provided up front in the main text.

Response: Thank you for this suggestion. In the revision, we ran MCMCtree in PAML to estimate the divergence time for each phylogenetic tree node, generated new **Fig. 2a** and added the relevant content in the main text.

R2-Q6: The authors use RELAX to discriminate genes with relaxed selection vs intensified, which is a nice approach. However, they do not carry through with their interpretation of these different classes of genes (e.g. paragraph beginning with line 261). Can they provide more of a distinction between genes with relaxed vs accelerated selection? They are all discussed together without distinction to their unique patterns of evolution, which muddles the discussion about the evolutionary trajectory of each of the genes.

Response: Thanks for this positive comment. We agree that the use of RELAX is a very nice approach. We also agree that it is difficult to interpret these patterns. However, we note that this difficulty is not unique to our study, even though papers do not always carefully explain these issues. Indeed, any pattern of elevated molecular evolution (e.g., elevated gene-wide dN/dS) in any study can be caused by either elevated positive selection at some sites, relaxed purifying selection at some sites, or a combination of the two. We used RELAX to identify signatures of gene-wide relaxation or intensification of selection. We also used RERconverge to identify signatures of gene-wide acceleration or deceleration protein evolution. These two approaches are complementary, but are not the same. For example, note that RELAX is focused specifically on detecting relaxation versus intensification of selection, while RERconverge is focused on identifying shifts in rates of molecular (i.e., protein) evolution more generally.

As we introduced (**R1-Q3**), relaxed selection (i.e., relaxation) is caused by weakening of both purifying selection ($\omega < 1$) and positive selection ($\omega > 1$), towards neutrality (see the arrow in **Additional figure 1**). Intensified selection (i.e., intensification) is caused by strengthening of both purifying selection ($\omega < 1$) and positive selection ($\omega > 1$), away from neutrality (see the arrow in **Additional figure 1**). In other words, RELAX tests whether a gene is under relaxed or intensified **selection**. We also used the complementary RERconverge to identify genes experiencing accelerated evolution or decelerated evolution. Based on estimated relative evolutionary rates, RER (i.e., protein evolutionary rate), we defined genes with significant higher RER in social branches (Permutation $P < 0.05$) as genes experiencing accelerated evolution (i.e., rapidly evolving genes), genes with significant lower RER in social branches (Permutation $P < 0.05$) as genes experiencing decelerated evolution (i.e., slowly evolving genes). Accelerated

evolutionary rates (i.e., higher RER) can be caused by increased positive selection, relaxed purifying selection, or a combination of the two. Similarly, decelerated evolutionary rate can be caused by decreased positive selection, intensified purifying selection, or a combination of the two (**Additional figure 2**). Although the uses of RERconverge and RELAX are two complementary but distinct approaches to identify genes experiencing shifts in patterns of convergent molecular evolution, we did find a set of significantly overlapping genes identified by both analyses (**Fig. 2g**). These overlapping genes can help explain that the pattern of accelerated evolution of genes in social branches (i.e., with significantly higher RER) is often driven by relaxed selection (e.g. relaxed purifying selection). We have tried to further clarify these issues in the revised manuscript.

Additional figure 1. Schematic diagrams depicting the cases of a gene experiencing relaxation or intensification of selection.

Additional figure 2. Schematic diagrams depicting the cases of a gene under accelerated evolution or decelerated evolution in social branches.

Responses to comments of Reviewer 3

In their study, the authors sequenced and assembled many new transcriptomes from social spider species with the goal of understanding which genetic changes are associate with the switch from solitary to social life. They compared those social, semi-social, and solitary species across a phylogeny to identify genome-wide patterns of differential selection, gene-specific patterns of accelerated selection, and convergent amino acid substitutions associated with social life. They identified a global pattern of relaxed selection in social species. They also report a general acceleration due to relaxation of constraint on genes in pathways related to neurogenesis and behavior.

Generally, I find the paper to address an interesting subject that readers will appreciate. The approaches here had not previously been applied to social life, to my knowledge. Their contributions in just genomics is substantial, and their methods are generally sound. However, I present major concerns detailed below that need to be addressed such that all aspects are up to field standards in terms of statistics and presentation to the reader.

Response: Thank you very much for these positive comments.

R3-Q1. The inferred species tree would be best reported with branch support values in Fig S1 so that the reader can understand which clades are strongly supported. Possibly even in Figure 1, such as with dots on nodes over a particular support value. Branch support values could be bootstrap support or approximate likelihood ratio test (PhyML).

Response: This is a great point. In the revised manuscript, we generated a new genome-scale phylogenetic tree with branch support values and divergent time as new **Fig. 2a**. In detail, we inferred the ML tree using RAxML, and estimated the divergent time using MCMCtree in PAML package.

R3-Q2. When reporting the enriched GO terms for the RERconverge and RELAX analyses, they should be filtered by and reported with their adjusted p-values or false discovery rates (FDRs). An FDR higher than 0.05 can be used. Such an adjustment would affect Figures 2 and 3, and some of the manuscript text. Judging by Supp Tables 5, 6, 8, and 9, there are still some relevant categories that pass adjustment, and using FDRs could help focus conclusions on the finding with the best statistical support.

Response: Our RERconverge and RELAX analyses follow the guidelines of the authors of these popular packages, which have both been successfully used in many published papers, (e.g., Kolora *et al.*, 2021. *Science*; Daane *et al.*, 2021. *Current Biology*; Zou *et al.*, 2021. *Molecular Biology and Evolution*; Rubin *et al.*, 2019. *Philosophical Transactions of the Royal Society B*). Specifically, with respect to using adjusted *P* values: the authors of RERconverge demonstrate in detail that their new permutation approach (which combines simulated trait evolution across the study phylogeny with permutation) is actually more conservative, and better accounts for the observed tree topology and hierarchical nature of GO terms, than a standard *P* value adjustment (e.g., with the `p.adjust` R function) (see Saputra *et al.* 2021, *Molecular Biology and Evolution*; see <https://github.com/nclark-lab/RERconverge/blob/master/vignettes/PermutationWalkthrough.pdf>). Thus, we report permutation *P* values, as recommended by the RERconverge package developers, and not nominal *P* values or adjusted *P* values, and no further adjustment is warranted.

Similarly, the authors of RELAX have used simulations and empirical datasets to assess the statistical power and performance of RELAX, and we have followed their guidelines (see Wertheim et al. 2014, *Molecular Biology and Evolution*). Specifically, using the nominal $P = 0.05$ in RELAX, which controls the false positive rate, as shown by Wertheim et al. (2014), the “..estimated type I error rate at 0.052 (95% confidence interval [CI]: 0.040–0.068; Wilson’s method).” Thus, no further P value adjustment is warranted (Liang et al., 2021, *Current Biology*; Chark et al., 2021, *Molecular Biology and Evolution*; Sun et al., 2021, *Molecular Biology and Evolution*; Cui et al., 2021, *Molecular Ecology*; Cui et al., 2021, *Cell*).

For GO enrichment of genes showing significant relaxation or intensification of selection from RELAX, we used the R package, topGO (<https://rdr.io/bioc/topGO/>) (Alexa and Rahnenfuhrer 2021) to perform gene term enrichment analysis (<https://rdr.io/bioc/topGO/f/inst/doc/topGO.pdf>). The package developers specifically recommend *not* to attempt any type of FDR/FWER adjustment procedure because they tend to be overly conservative. topGO is a very popular GO enrichment package, and indeed none of the papers we looked at (e.g., Mantri et al., 2021. *Nature Communications*; Carey et al., 2021. *Science Advances*; Layton et al., 2021. *Nature Climate Change*) used any subsequent P -adjustment procedure. We have followed the advice of the topGO developers and other researchers using topGO.

R3-Q3. Positive comment. The significant overlap between the RELAX and RERconverge results nicely bolsters the authors conclusions.

Response: Thanks very much for this positive comment. In the revision, we moved the Venn diagram showing this overlap to the main text as a new **Fig. 3g**. We do note, as described above, that RELAX and RERconverge are complementary but distinct.

R3-Q4. The TPC1 study. Could the authors comment on where TPC1 is expressed? Neuronal? Brain? Is it known to affect behavior or social behavior in arthropods?

Response: This is an interesting question. We retrieved the tissue-specific transcriptome data from one social spider species, *Stegodyphus dumicola* (Tong et al., 2020, *Genome Biology and Evolution*), and found that *TPC1* is broadly expressed in the brain, venom gland, leg, and

abdomen. However, after re-running FADE with amino acid alignments from the improved transcriptome and genome assemblies, we found that *TPC1* did not experience convergent substitutions across social branches.

R3-Q5. Searching for convergent amino acid substitutions out of the whole genome is notoriously difficult. Does FADE provide a way of understanding how many apparently convergent changes would be expected by chance? Did the authors see a relative excess compared to that expectation?

Response: We appreciate your understanding about the difficulties in identifying the convergent amino acid substitutions. This is a great and difficult question. As suggested by your question, although there are many tools aiming to detect the convergent amino acid site, there is no ideal one that works perfectly given our relatively large phylogeny and study question focused on detecting signatures of molecular convergence across an estimated eight independent origins of sociality. In this scenario, we might expect that there would be zero substitutions only found on social branches and found across all social branches. FADE was recently developed to test for directional selection at individual sites in a protein alignment (<https://hyphy.org/tutorials/CL-prompt-tutorial>), and we attempted to take advantage of this new tool in our study to identify the convergent amino acid substitutions. FADE does not explicitly report the number of expected convergent substitutions that would be expected by chance. Instead, FADE tests for each site in the alignment whether there is a substitution bias towards a specific amino acid in the foreground branches (i.e., social branches) relative to background branches (i.e., nonsocial branches). High values of the bias parameter indicate that the site has experienced more than expected substitutions towards a particular amino acid. The bias parameter is reported as a Bayes Factor (BF), where $BF > 100$ is taken to provide strong evidence that the site is evolving under directional selection (<https://github.com/veg/hyphy-site/blob/master/docs/methods/selection-methods.md>).

REVIEWER COMMENTS

Reviewer #1 (Remarks to the Author):

This article seeks convergences in patterns of genome evolution across social spider species with 7-8 independent origins of sociality. By examining many more independent and recent origins of sociality than have been studied in any previous study, this work greatly advances our understanding about the consequences of social evolution on genome-wide selection. I find the revision to be greatly improved in terms of clarity and flow. I appreciate the authors' careful responses to all reviewer points from the previous round of reviews. I noticed just a few small mistakes in my reading that could be addressed, but otherwise the article looks great to me.

Minor suggestions:

Lines 264-265: There are two citations that are written out instead of being numbered.

Lines 354-355: States that there are eight branches that independently evolved sociality. This is slightly confusing throughout the manuscript, as the tree in figure 2a (with 8 independently origins) does not match the tree in figure S3 (with 7 independent origins).

Reviewer #2 (Remarks to the Author):

The manuscript has been substantially revised and improved according to reviewer suggestions, including removing questionable datasets, reanalyzing large portions of the data, and clarifying several components of the biology and interpretation. Given this thorough revision, I do not have any additional comments. This is novel work on a very interesting system for understanding social biology.

Reviewer #3 (Remarks to the Author):

My original major concern was not addressed. To state it again, R3-Q2 "When reporting the enriched GO terms for the RERconverge and RELAX analyses, they should be filtered by and reported with their adjusted p-values or false discovery rates (FDRs). An FDR higher than 0.05 can be used. Such an adjustment would affect Figures 2 and 3, and some of the manuscript text. Judging by Supp Tables 5, 6, 8, and 9, there are still some relevant categories that pass adjustment, and using FDRs could help focus conclusions on the finding with the best statistical support."

The authors rebut with claims that p-value adjustment for multiple hypothesis testing is not needed for any of the methods they employed. Below, I present arguments for each that adjustment is indeed necessary.

For RERconverge: The Permutation paper (see Saputra et al. 2021, Molecular Biology and Evolution) describes permutations/simulations that produce a phylogenetically-aware null distribution that is then

used to calculate an empirical p-value to replace the parametric p-values that they used before. The permutations correct for phylogenetic nonindependence (a better null) and for correlations between the genes in an annotated functional group, but neither of these removes the necessity of subsequent correction for multiple hypothesis testing. In fact, the permutation authors apply p-value adjustments consistently after calculating these empirical p-values. For example, in their study of binary phenotypes, they state that “The resulting P values were corrected for multiple hypothesis testing using Storey’s correction”. Also the functional enrichments in Table 1 are only reported after p-value adjustment by the Benjamini-Hochberg method, for both parametric and empirical (permulated) methods.

The authors state that the RELAX method controls false positive rate well. That is good, but false positive rate is a distinct problem from multiple hypothesis testing. It doesn’t obviate the need for corrections for multiple hypothesis testing. RELAX isn’t even aware of how many genes are being analyzed serially, and it certainly doesn’t know how many functional annotations will be later tested in a GO term enrichment program.

For the GO enrichments, the TopGO instruction manual, as cited in the authors’ rebuttal, is not a peer-reviewed article. The 3 arguments presented in section 6.2 of that instruction manual are not convincing and do not provide supporting evidence for not performing p-value adjustment for multiple hypothesis testing. To their first point, just because an analysis had no enrichments that pass adjustment doesn’t mean the user can skip adjustment because they were disappointed. What that result means is that the study was underpowered or there is no true signal. For the second point, even if an adjustment procedure could be improved to fit a particular dataset, that doesn’t obviate the necessity of performing some kind of adjustment for multiple hypothesis testing. Third point, there is no supporting evidence that elim and weight account for multiple testing. They adjust for the local structure of the DAG. Also, their Bioinformatics paper states their ‘elim’ and ‘weight’ methods control false-positive rate, but false-positive rate is a different issue from the problem of multiple hypothesis testing, and they don’t provide supporting evidence that it addresses multiple hypothesis testing or really even claim that it does in their Bioinformatics paper (2006).

If the authors of the topGO instructional manual want to claim that multiple hypothesis testing is not needed, then they need to publish their evidence for that. If they have done that, then will the authors please provide that reference? I understand that statistical procedures are not yet perfect, but we must still apply the best that we have. Otherwise, the field of genomics will make unreliable conclusions due to the study of noise in our data.

Author Response to Reviewer Comments:

Thank you very much for the opportunity for a second round of revision for our manuscript NCOMMS-21-10286A to Nature Communications. We thank the three Reviewers and the Associate Editor for their constructive comments. We agree with all the Reviewer comments and have carefully revised our manuscript as they suggest. Most importantly, as argued by Reviewer 3, in the revised manuscript, we appropriately correct all of our results for multiple hypothesis testing. We believe that, as a result of this second round of revisions, our revised manuscript is further greatly improved.

We address all of the Reviewer comments and suggestions in the detailed point-by-point list below:

Responses to comments of Reviewer 1

RI: This article seeks convergences in patterns of genome evolution across social spider species with 7-8 independent origins of sociality. By examining many more independent and recent origins of sociality than have been studied in any previous study, this work greatly advances our understanding about the consequences of social evolution on genome-wide selection. I find the revision to be greatly improved in terms of clarity and flow. I appreciate the authors' careful responses to all reviewer points from the previous round of reviews. I noticed just a few small mistakes in my reading that could be addressed, but otherwise the article looks great to me.

Response: Thank you very much for your very positive comments on our previous revision. We agree that this comparative genomics study greatly advances our understanding of the genic and genome-wide consequences of social evolution.

RI-Q1: Lines 264-265: There are two citations that are written out instead of being numbered.

Response: We apologize for this citation format issue. We now correct it in the revision.

RI-Q2: Lines 354-355: States that there are eight branches that independently evolved sociality. This is slightly confusing throughout the manuscript, as the tree in figure 2a (with 8 independent origins) does not match the tree in figure S3 (with 7 independent origins).

Response: Thanks for pointing out this typo. We now correct the figure S3 with 8 independent origins.

Responses to comments of Reviewer 2

R2: The manuscript has been substantially revised and improved according to reviewer suggestions, including removing questionable datasets, reanalyzing large portions of the data, and clarifying several components of the

biology and interpretation. Given this thorough revision, I do not have any additional comments. This is novel work on a very interesting system for understanding social biology.

Response: Thank you for your very positive comments. We agree that spiders are an excellent study system for understanding social evolution.

Responses to comments of Reviewer 3

R3-Q1: My original major concern was not addressed. To state it again, R3-Q2 “When reporting the enriched GO terms for the RERconverge and RELAX analyses, they should be filtered by and reported with their adjusted p-values or false discovery rates (FDRs). An FDR higher than 0.05 can be used. Such an adjustment would affect Figures 2 and 3, and some of the manuscript text. Judging by Supp Tables 5, 6, 8, and 9, there are still some relevant categories that pass adjustment, and using FDRs could help focus conclusions on the finding with the best statistical support.”

The authors rebut with claims that p-value adjustment for multiple hypothesis testing is not needed for any of the methods they employed. Below, I present arguments for each that adjustment is indeed necessary.

For RERconverge: The Permutation paper (see Saputra et al. 2021, Molecular Biology and Evolution) describes permutations/simulations that produce a phylogenetically-aware null distribution that is then used to calculate an empirical p-value to replace the parametric p-values that they used before. The permutations correct for phylogenetic nonindependence (a better null) and for correlations between the genes in an annotated functional group, but neither of these removes the necessity of subsequent correction for multiple hypothesis testing. In fact, the permutation authors apply p-value adjustments consistently after calculating these empirical p-values. For example, in their study of binary phenotypes, they state that “The resulting P values were corrected for multiple hypothesis testing using Storey’s correction”. Also the functional enrichments in Table 1 are only reported after p-value adjustment by the Benjamini-Hochberg method, for both parametric and empirical (permulated) methods.

The authors state that the RELAX method controls false positive rate well. That is good, but false positive rate is a distinct problem from multiple hypothesis testing. It doesn’t obviate the need for corrections for multiple hypothesis testing. RELAX isn’t even aware of how many genes are being analyzed serially, and it certainly doesn’t know how many functional annotations will be later tested in a GO term enrichment program.

For the GO enrichments, the TopGO instruction manual, as cited in the authors’ rebuttal, is not a peer-reviewed article. The 3 arguments presented in section 6.2 of that instruction manual are not convincing and do not provide supporting evidence for not performing p-value adjustment for multiple hypothesis testing. To their first point, just because an analysis had no enrichments that pass adjustment doesn’t mean the user can skip adjustment because they were disappointed. What that result means is that the study was underpowered or there is no true signal. For the second point, even if an adjustment procedure could be improved to fit a particular dataset, that doesn’t obviate the necessity of performing some kind of adjustment for multiple hypothesis testing. Third point, there is no supporting

evidence that elim and weight account for multiple testing. They adjust for the local structure of the DAG. Also, their Bioinformatics paper states their 'elim' and 'weight' methods control false-positive rate, but false-positive rate is a different issue from the problem of multiple hypothesis testing, and they don't provide supporting evidence that it addresses multiple hypothesis testing or really even claim that it does in their Bioinformatics paper (2006).

If the authors of the topGO instructional manual want to claim that multiple hypothesis testing is not needed, then they need to publish their evidence for that. If they have done that, then will the authors please provide that reference? I understand that statistical procedures are not yet perfect, but we must still apply the best that we have. Otherwise, the field of genomics will make unreliable conclusions due to the study of noise in our data.

Response: Thank you very much for your detailed comments on the importance of appropriately correcting for multiple hypothesis testing and also for sharing your criticisms of topGO and the topGO manual. We are completely convinced. We had thought that we were correctly following the guidelines of the authors of the RERconverge, RELAX, and topGO packages, and also the approach of many previously published papers, but we were incorrect. We are happy to have the opportunity to fix this important issue. We also definitely agree with your statement that the fields of comparative genomics, transcriptomics, etc. would be greatly improved if all researchers appropriately corrected their results for multiple comparisons. In the revised manuscript, we correct for multiple comparisons for each of the analyses: RERconverge, RELAX and associated gene ontology (GO) enrichment, using a cutoff false discovery rate (FDR) of 15%, following a recent study (Partha *et al.*, *eLife*, 2017). Specifically, as described in detail in the revised manuscript, for RERconverge, we estimate q-values (using the R package qvalue) for each gene and GO term from the permutation p-values (using 10,000 permutations) output from RERconverge; similarly, we estimate q-values for each gene from the p-values output from RELAX, and we similarly estimate q-values following GO enrichment. We agree with the criticisms of topGO, and in the revised manuscript, we use GOATOOLS (Klopfenstein *et al.*, *Scientific Reports*, 2018) for GO enrichment, which has been widely used in recent comparative genomics studies (e.g., Xu *et al.*, *Molecular Biology and Evolution*, 2021; Mesny *et al.*, *Nature Communications*, 2021; Yang *et al.*, *BMC Biology*, 2021). We used eggNOG-mapper (<http://eggno-mapper.embl.de/>) in the revised manuscript to annotate orthologous genes with assigned GO terms.

As expected, when we appropriately corrected for multiple comparisons using q-values with an FDR cutoff for significance of 0.15, the number of genes and enriched GO terms identified by RERconverge and RELAX did decrease when compared to the previous version of our manuscript. However, as shown in the revised manuscript, the main results and biological insights of our study remain, and we agree with the Reviewer that our revised results are much more robust.

REVIEWERS' COMMENTS

Reviewer #3 (Remarks to the Author):

The authors have responded with new statistical analyses that are sophisticated and robust. There are no more concerns about the study and I recommend publication. I also thank the authors for their collegial response and congratulate them on an interesting study.